# DRDFL: Divide-and-conquer Collaboration for Efficient Ring-topology Decentralized Federated Learning

## Abstract

Federated learning traditionally relies on server-based architecture, which often incur high communication costs and suffer from single points of failure. To avoid these limitations, we explore Ring-topology Decentralized Federated Learning (RDFL), a fully decentralized paradigm that enables peer-to-peer training. However, the inherent challenge of data heterogeneity is further amplified in RDFL due to limited communication bandwidth cross clients and the sparse connectivity of the ring topology. In this paper, we propose the Divide-and-conquer collaboration RDFL framework (DRDFL), which captures underlying data patterns by jointly learning personalized and invariant knowledge through two complementary modules with distinct optimization objectives. Specifically, each client trains a transferable *Learngene* module via adversarial optimization against a uniform label distribution to learn consensus knowledge, thereby mitigating label distribution skew induced by data heterogeneity. To simultaneously alleviate feature distribution skew, a personalized *PersonaNet* module is introduced that models local features using a Gaussian mixture distribution and updates them based on the global class representation. Clients only share lightweight *Learngene* and global representations with a directed neighbor, which guarantees flexible choices for resource efficiency and better convergence. Extensive experiments show that our method achieves superior performance in RDFL while reducing the communication cost to only 0.58 M, which is more than two orders of magnitude lower than the state-of-the-art baseline. This substantial reduction highlights the effectiveness of our approach in addressing data heterogeneity under stringent communication constraints.

## 1 Introduction

Federated learning (FL) is a distributed learning paradigm that allows multiple clients to collaboratively train a global model while keeping data local (McMahan et al., 2017; Xue et al., 2025; Qi et al., 2023). One major challenge of FL is data heterogeneity, caused by distributional differences across clients (Albshaier et al., 2025; Yang et al., 2024; Li et al., 2024b; Qi et al., 2025). Recent works addressing this challenge mainly focus on the centralized FL (CFL) setting, where a central server orchestrates the learning among clients and is responsible for parameter aggregation after receiving locally trained models on the edge. In practice, the server may experience system failures or malicious attacks, potentially leading to privacy leakage or interruptions in training. Moreover, since all communication flows through the server, it becomes a bottleneck and incurs substantial bandwidth overhead (Li et al., 2024c).

With this regard, decentralized FL (DFL) has recently emerged as a promising method for reducing the communication bandwidth of the busiest node and embracing peer-to-peer communication for faster convergence (Dai et al., 2022). In DFL, no global model state exists, the participating clients follow a communication protocol to reach a so-called consensus model. Classical fully-connected or dynamically-varying FL architectures typically assume dense client connectivity, which results in excessive communication overhead and severely limits their scalability in large-scale real-world scenarios (Zhang et al., 2024; Li et al., 2025b). Ring-topology Decentralized Federated Learning (RDFL) (Li et al., 2023) restricts interactions to local neighbors, thereby minimizing redundant

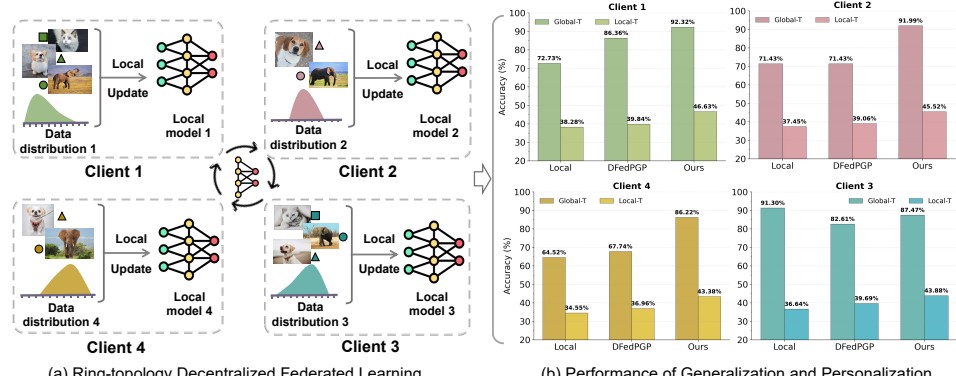

Figure 1: Illustration of the optimized learning mechanism in RDFL and comparative performance of the proposed method against baselines in terms of generalization and personalization. **(a)** Depicts non-IID data heterogeneity in a decentralized federated learning setting with a ring topology, where shared models are collaboratively optimized through client interactions. **(b)** Compares the personalization (Local-T) and generalization (Global-T) performance of local models collaboratively trained across four clients with heterogeneous data distributions.

transmissions, as shown in Figure 1 (a). This design has demonstrated promising progress in peer-to-peer applications such as collaborative autonomous driving using vehicle-to-vehicle networks and edge IoT systems (Nguyen et al., 2022; Yuan et al., 2024; Li et al., 2025a), underscoring its practical relevance.

Nevertheless, our exploration of RDFL reveals that it still suffers from intrinsic communication constraints that hinder efficient information exchange, thereby exacerbating data heterogeneity across clients. We revisit this issue by analyzing the underlying sources of heterogeneity in RDFL and identify two primary forms of distribution shift: **feature distribution skew**, where identical classes exhibit differing feature patterns across clients due to varying local contexts, and **label distribution skew**, where class frequency distributions vary significantly between clients. These skews pose distinct challenges:

*Feature skew undermines representation consistency, whereas label skew leads to biased updates and poor generalization to underrepresented classes.*

Most existing DFL methods predominantly focus on addressing feature distribution skew, aiming to improve personalized performance through strategies such as partial model adaptation and sparse parameter masking Kairouz et al. (2021); Li et al. (2022); Dai et al. (2022); Liu et al. (2024). While these techniques effectively capture client-specific representations, they often skip the equally critical problem of label distribution skew across clients and the inherent bandwidth constraints imposed by RDFL. In contrast to centralized FL, RDFL lacks a global server to facilitate consistent aggregation, requiring the consensus knowledge learned collaboratively among clients to be more generalizable. As illustrated in Figure 1 (b), methods such as *Local* and DFedPGP (Liu et al., 2024) demonstrate outstanding personalized performance (Local-T) but exhibit limited generalization capability (Global-T) to the global data distribution. In contrast, our approach enhances the model's generalization ability toward the global data distribution while also affecting its personalized performance. This highlights the importance of simultaneously addressing both types of distribution skew in limited-communication RDFL, as well as the necessity of training generalized and effective consensus knowledge.

Inspired by the recently proposed Learngene paradigm [1] (Wang et al., 2022a; Feng et al., 2025; Wang et al., 2023; Xia et al., 2024b), which encapsulates consensus knowledge within the lightweight model to facilitate efficient task adaptation, we propose a **D**ivide-and-conquer collaboration **R**ing-topology **D**ecentralized **F**ederated **L**earning (DRDFL) method. To address the challenge of label distribution skew across clients, we devised a transferable *Learngene* module that undergoes adversarial optimization under a uniform label distribution constraint. This facilitates the learning of unbiased

---

[1]"Learngene" refers to the machine learning paradigm, while *Learngene* denotes the specific model component instantiated in our framework.

representations that are independent of the client and class invariance. In parallel, to mitigate feature distribution skew, we introduce a personalized *PersonaNet* for each client. This module captures client-specific feature semantics by modeling local class features using the Gaussian mixture distribution while aligning with the global class statistics. This allows *PersonaNet* to learn representations that preserve local discriminative patterns while remaining semantically aligned with the global context. During reconstruction, invariant features from *Learngene* are fused with personalized features from *PersonaNet* and passed through a decoder for input reconstruction. Gaussian noise is injected into the reconstructed data to promote robust classifier training and prevent overfitting. In RDFL, the consensus *Learngene* and class distributions are iteratively optimized and cyclically shared among clients, enabling effective collaborative learning to accelerate convergence. Our contributions are summarized as follows:

- We revisit data heterogeneity in RDFL, where limited communication and the absence of a central coordinator amplify its impact, and reveal the importance of simultaneously considering both distribution skewness issues and the necessity of training generalized consensus knowledge.

- We propose a novel framework, Divide-and-conquer collaboration RDFL, to address these challenges by introducing a consensus *Learngene* module through adversarial optimization training, and a personalized *PersonaNet* module optimized for Gaussian mixture consistency.

- Extensive experiments against 8 state-of-the-art baselines demonstrate that DRDFL attains comparable generalization to centralized FL while delivering superior personalization over existing decentralized methods. Remarkably, this performance is achieved with only **0.58 M** communicated parameters, which is much smaller than advanced methods.

## 2 RELATED WORK

**Federated Learning** (FL) has emerged as a promising paradigm for privacy-preserving machine learning (McMahan et al., 2017). To avoid single points of failure, decentralized federated learning (DFL) has gained traction, where clients interact with neighbors via point-to-point communication to collaboratively train models without relying on a central server. Recent efforts in personalized DFL have explored various optimization and model adaptation strategies. DisPFL (Dai et al., 2022) designs client-specific models and pruning masks to accelerate convergence, while KD-PDFL (Liu et al., 2022) applies knowledge distillation to capture statistical differences across clients. ARDM (Sun et al., 2022) establishes theoretical lower bounds for communication and computation costs, and DFedPGP (Liu et al., 2024) leverages multi-step updates with alternating optimization to improve stability and convergence. In addition, DFML (Khalil et al., 2024) mitigates the drift toward local objectives by applying a re-weighted SoftMax loss (Legate et al., 2023). While showing promising results on personalization, the model exhibits inferior generalization performance, possibly due to the limited scalability of the input parameters. DRDFL can adapt intermediate features, enhancing generalization and providing greater flexibility in handling diverse data distributions.

**Disentangled Representation Learning** aims to uncover and separate the underlying factors of variation in data, thereby improving model generalization and interpretability (Wang et al., 2024b; Zhu et al., 2021; Guo et al., 2024b). Variational Autoencoders (VAEs) (Kingma, 2013) provide a principled framework for learning such representations by maximizing the evidence lower bound (ELBO): $\log p(\boldsymbol{x}) \geq \mathbb{E}q\phi(\mathbf{z}|\boldsymbol{x})[\log p_\theta(\boldsymbol{x}|\mathbf{z})] - D_{\mathrm{KL}}(q_\phi(\mathbf{z}|\boldsymbol{x})|p(\mathbf{z}))$, where the first term corresponds to the reconstruction objective and the second enforces alignment between the posterior and a standard Gaussian prior. Conditional VAE (CVAE) (Sohn et al., 2015) further incorporates label information into both the encoder and decoder to improve semantic consistency and alleviate latent collapse. Recent studies have applied disentangled learning to CFL to address data heterogeneity (Yan & Long, 2023; Luo et al., 2022; Chen & Zhang, 2024; Wu et al., 2024). These works introduce mechanisms such as invariant aggregation, gating strategies, and orthogonal decomposition to separate shared and personalized components. However, most rely on centralized server-side coordination or proxy datasets, which limits their applicability to DFL. In contrast, our approach operates entirely in a serverless setting, enhancing communication efficiency and consistency through fully decentralized client interactions while preserving the benefits of disentangled representation learning.

**Learngene** (Lin et al., 2024; Xia et al., 2024b;a; Li et al., 2024a; Xie et al., 2025; Wang et al., 2023), a novel paradigm of machine learning inspired by biological genetics, has been proposed to

condense a large-scale ancestral model into generalized *Learngene* that adaptively initialize models for various downstream tasks. Wang et al. (2022a) first introduced Learngene based on the gradient information of ancestral models and demonstrated its effectiveness in initializing new task models in open-world scenarios, thereby reflecting its strong generalization capability. To rapidly construct a diverse variety of networks with varying levels of complexity and performance trade-offs, the customized Learngene pool (Shi et al., 2024) methodology is tailored to meet resource-constrained environments. Furthermore, Feng et al. (2024) further validated that transferring core knowledge through *Learngene* is both sufficient and effective for neural networks. Inspired by this, we propose to transfer an encapsulating consensus knowledge *Learngene* module across clients, offering a novel perspective for collaborative knowledge sharing in RDFL and enabling each client to learn from others in a decentralized manner.

## 3 PRELIMINARIES

### 3.1 PROBLEM FORMULATION

The Ring-topology Decentralized Federated Learning (RDFL), as one of the most representative and lightweight sparse structures in partially connected decentralized FL, aims to enable efficient distributed learning across multiple data sources under privacy constraints. Compared to fully connected networks, RDFL significantly reduces communication overhead by restricting interactions to local neighbors (Wang et al., 2022b; Beltrán et al., 2023; Wang et al., 2024a). However, such sparsity also amplifies the negative impact of data heterogeneity, since each client can only exchange information with its immediate peers. Given these considerations, we revisit and further refine the challenges of data heterogeneity within the RDFL architecture, specifically including:

**Definition 3.1** (Feature Distribution Skew). *Let $p_i(\boldsymbol{x})$ and $p_j(\boldsymbol{x})$ represent the feature distributions for train sample $i$ and test sample $j$, respectively. The feature distribution of the training samples may be different from that of the test samples, but the class conditional distribution of the same class remains invariant, i.e.,*

$$p_i(\boldsymbol{x}) \neq p_j(\boldsymbol{x}) \quad \text{but} \quad p_i(y|\boldsymbol{x}) = p_j(y|\boldsymbol{x}).$$

**Definition 3.2** (Label Distribution Skew). *Let $p_i(y)$ and $p_j(y)$ denote the label distributions for client $i$ and client $j$, respectively. The label distributions across clients may differ, but the class-conditional feature distributions remain invariant, i.e.,*

$$p_i(\boldsymbol{x}|y) = p_j(\boldsymbol{x}|y) \quad \text{but} \quad p_i(y) \neq p_j(y).$$

### 3.2 DIVIDE-AND-CONQUER COLLABORATION

**Maximizing the learning of consensus knowledge while fitting class-specific distributions is reasonable to mitigate feature distribution skew.** Each client's local training and test datasets can be generated in different contexts/environments. For example, a client's training image samples may be primarily captured by a local camera, while the test images may come from the Internet and have different styles. From a contextual perspective, the target learning model must have a certain level of generalization capability to perform well in unknown and diverse contexts. We propose to train the *PersonaNet* module based on the global class mean and variance derived from collaborative learning, allowing the capture of personalized information while mitigating feature distribution skew.

**Training with a uniform prior distribution provides a principled solution to label distribution skew.** Due to the inherent limitations of local clients, which are limited to their specific data subsets, they often fail to adequately represent the broader data distribution. Consequently, must collaborate to overcome the bottlenecks imposed by limited individual datasets. We emphasize the use of adversarial classifiers in training the *Learngene* module within the RDFL system to adapt to a unified prior distribution $p_u(y = k) = 1/K$, where $K$ represents the total number of classes. This ensures that cross-client collaboration is not affected by inconsistencies in class distributions, promoting the learning of a stable and invariant latent space, improving the generalization capability of the model.

## 4 METHODOLOGY

### 4.1 NOTATIONS

Consider a typical setting of RDFL with $M$ clients, each client $m$ has a dataset $\mathcal{D}_m = \{(\boldsymbol{x}_i, y_i)\}_{i=1}^{|\mathcal{D}_m|}$, where $y_i \in [1, K]$ and $K$ is the number of overall classes. The optimization problem that RDFL to

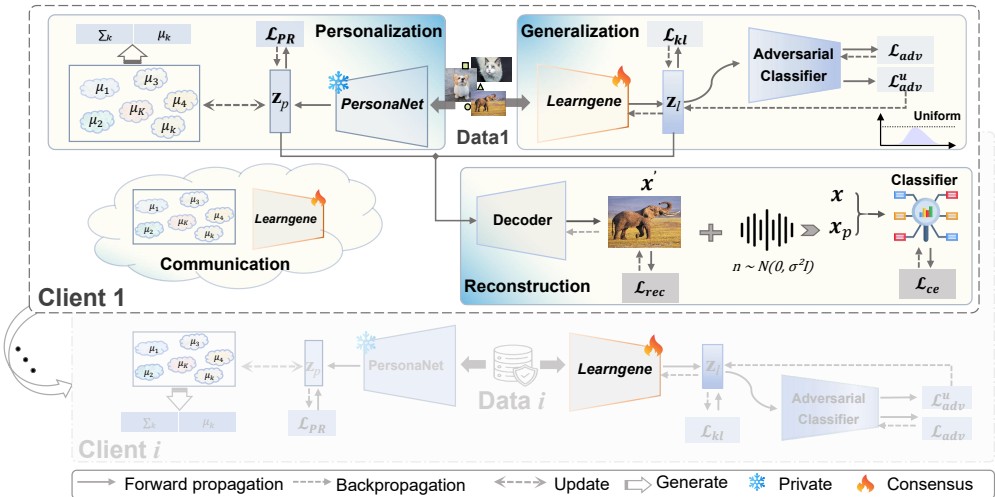

Figure 2: Overview of the DRDFL framework. The *PersonaNet* module learns class-specific personalized representations via a Gaussian mixture to enhance local adaptability, while the consensus module *Learngene* is optimized via adversarial training against a uniform label distribution to capture globally invariant knowledge and enhance cross-client consistency. During reconstruction, noise is injected into reconstructed data to improve classifier robustness. Global latent Gaussian representations and consensus *Learngene* are cyclically updated across the topology to enable collaborative learning.

solve can be formulated as:

$$\min_{\boldsymbol{w}_m} \mathcal{L}(\boldsymbol{w}_m) = \frac{1}{|\mathcal{D}_m|} \sum_i \ell\left(\boldsymbol{x}_i, y_i; \boldsymbol{w}_m\right), \tag{1}$$

where $\boldsymbol{w}_m$ is the model parameter and $\mathcal{L}(\boldsymbol{w}_m)$ is the empirical risk computed from $m$-th client data $\mathcal{D}_m$, and $\ell$ is a loss function applied to each data instance.

In the serverless DRDFL framework, the underlying goal of training a model with both personalized and generalized capabilities can be specifically described as: (1) identifying highly discriminative class-specific attributes to ensure accurate classification, and (2) mining class-independent common attributes to enhance the model's generalization ability. As illustrated in Figure 2, we introduce a divide-and-conquer collaboration mechanism inspired by the variational autoencoder (VAE) to achieve this objective. Specifically, we design two complementary modules: a personalization module (*PersonaNet*, parameterized by $\psi_m$) for extracting client-specific representations, and a consensus generalization module (*Learngene*, parameterized by $\phi$) for capturing globally shared knowledge through collaborative learning across clients. The decoder module $p_{\theta_m}$ integrates the outputs of both branches to reconstruct the input data, which is then perturbed with noise and passed to the classifier $f_{\omega_m}$ (parameterized by $\omega_m$) for robust training. Consequently, each local model is structured as $\boldsymbol{w}_m = [\psi_m, \phi, \theta_m, \omega_m]$, following a divide-and-conquer strategy, where $\phi$ is used for the cross-client consensus module and other modules are private. For simplicity, we unify them without subscripts and focus on training a model with the dual optimization objectives of generalization and personalization. The optimization of $[\psi, \phi, \theta]$ is achieved by maximizing the ELBO to provide a tight lower bound for the original $\log p(\boldsymbol{x})$:

$$
\begin{aligned}
\max_{\psi,\phi,\theta} \mathbb{E}_{\boldsymbol{x}} \big[ \mathbb{E}_{q_\psi(\mathbf{z}_p, k|\boldsymbol{x}), q_\phi(\mathbf{z}_l|\boldsymbol{x})} \left[\log p_\theta(\boldsymbol{x}|\mathbf{z}_p, \mathbf{z}_l)\right] \\
- \underbrace{D_{\mathrm{KL}}\left(q_\psi(\mathbf{z}_p, k|\boldsymbol{x}) \| p(\mathbf{z}_p, k)\right)}_{PersonaNet} \\
- \underbrace{D_{\mathrm{KL}}\left(q_\phi(\mathbf{z}_l|\boldsymbol{x}) \| p(\mathbf{z}_l)\right)}_{Learngene}\big],
\end{aligned}
\tag{2}
$$

where ***first*** term represents the negative reconstruction error. The ***PersonaNet*** term enforces $q_\psi(\mathbf{z}_p, k|\boldsymbol{x})$ to align with the global class-specific prior Gaussian distribution, encouraging *PersonaNet* to generate latent representations with strong class discriminability. The ***Learngene*** term

promotes the alignment of latent representations generated by *Learngene* with the standard multivariate normal prior $p(\mathbf{z}_l)$, enabling the extraction of class-invariant information across clients.

## 4.2 PERSONALIZED *PersonaNet* TRAINING VIA GAUSSIAN MIXTURE DISTRIBUTION

The goal of the *PersonaNet* module is to ensure model personalization to mitigate feature distribution skew. Based on the large-margin Gaussian mixture loss (Wan et al., 2018; Zheng & Sun, 2019), we assume that the latent code $\mathbf{z}_p$ learned from the training set follows a Gaussian mixture distribution expressed as:

$$p(\mathbf{z}_p) = \sum_k \mathcal{N}(\mathbf{z}_p; \boldsymbol{\mu}_k, \boldsymbol{\Sigma}_k) p(k), \tag{3}$$

where $\boldsymbol{\mu}_k$ and $\boldsymbol{\Sigma}_k$ represent the mean and covariance of class $k$ in the feature space, and $p(k)$ denotes the prior probability of class $k$. Under this assumption, we encourage $\mathbf{z}_p$ to capture the necessary information related to the class label $y$.

Given a class label $y \in [1, K]$, the conditional probability distribution of $\mathbf{z}_p$ is defined as $p(\mathbf{z}_p|y) = \mathcal{N}(\mathbf{z}_p; \boldsymbol{\mu}_y, \boldsymbol{\Sigma}_y)$. Therefore, the corresponding posterior probability distribution is formulated as:

$$p(y|\mathbf{z}_p) = \frac{\mathcal{N}(\mathbf{z}_p; \boldsymbol{\mu}_y, \boldsymbol{\Sigma}_y) p(y)}{\sum_{k=1}^{K} \mathcal{N}(\boldsymbol{x}; \boldsymbol{\mu}_k, \boldsymbol{\Sigma}_k) p(k)}. \tag{4}$$

Then maximizing the mutual information between $\mathbf{z}_p$ and $k$ is transformed into calculating the cross entropy between the posterior probability distribution and the one-hot encoded class label:

$$\begin{aligned}
\mathcal{L}_{cls} &= -\sum_{k=1}^{K} \mathbb{I}(y = k) \log q(k|\mathbf{z}_p) \\
&= -\log \frac{\mathcal{N}(\mathbf{z}_p; \boldsymbol{\mu}_y, \boldsymbol{\Sigma}_y) p(y)}{\sum_{k=1}^{K} \mathcal{N}(\boldsymbol{x}; \boldsymbol{\mu}_k, \boldsymbol{\Sigma}_k) p(k)},
\end{aligned} \tag{5}$$

where the indicator function $\mathbb{I}(\cdot)$ equals 1 if $y$ is equal to $k$, and 0 otherwise. Here, $q(k|\mathbf{z}_p)$ refers to the auxiliary distribution introduced to approximate $p(k|\mathbf{z}_p)$, since directly optimizing $p(k|\mathbf{z}_p)$ is challenging in practice, as discussed in InfoGAN (Chen et al., 2016).

Recall that in *PersonaNet* term of Eq. 2 the Kullback-Leibler (KL) divergence between $q_\psi(\mathbf{z}_p, k|\boldsymbol{x})$ and $p(\mathbf{z}_p, k)$ is minimized. If the covariance matrix of $p(\mathbf{z}_p|y)$ tends to zero, then the distribution tends to a degenerate Gaussian distribution, is expressed as $p(\mathbf{z}_p|y) \rightarrow \delta(\mathbf{z}_p - \boldsymbol{\mu}_y)$. That is, all samples tend to the class mean $\boldsymbol{\mu}_y$. The KL divergence term degenerates into negative log-likelihood:

$$\mathcal{L}_{log} = -\log \mathcal{N}(\mathbf{z}_p; \boldsymbol{\mu}_y, \boldsymbol{\Sigma}_y), \tag{6}$$

where $\mathbf{z}_p$ denotes the mean output from the *PersonaNet*. The $\boldsymbol{\mu}_y$ and $\boldsymbol{\Sigma}_y$ dynamically updated using an EMA strategy, i.e., $\boldsymbol{\mu}_y = \alpha \boldsymbol{\mu}_y + (1-\alpha)\tilde{\boldsymbol{\mu}}_y$, $\boldsymbol{\Sigma}_y = \alpha \boldsymbol{\Sigma}_y + (1-\alpha)\tilde{\boldsymbol{\Sigma}}_y$, where $(\tilde{\boldsymbol{\mu}}_y, \tilde{\boldsymbol{\Sigma}}_y)$ denote the globally shared Gaussian statistics received from neighboring clients. The total loss for *PersonaNet* is given by: $\mathcal{L}_{PR} = \mathcal{L}_{cls} + \mathcal{L}_{log}$.

## 4.3 GENERALIZED *Learngene* TRAINING WITH ADVERSARIAL CLASSIFIER

Intuitively, we aim to decompose the latent space $\mathbf{z}$ such that $\mathbf{z}_l$ follows to a fixed prior distribution associated with knowledge shared across classes, independent of labels. This ensures that the resulting *Learngene* encoding module possesses the advantage of being inheritable and transferable. Specially, the *Learngene* term of in Eq. 2 is implemented by minimizing the KL divergence between $q_\phi(\mathbf{z}_l|\boldsymbol{x})$ and the prior $p(\mathbf{z}_l)$:

$$\mathcal{L}_{kl} = D_{\text{KL}}[q_\phi(\mathbf{z}_l|\boldsymbol{x}) \| p(\mathbf{z}_l)] = D_{\text{KL}}[\mathcal{N}(\boldsymbol{\mu}, \boldsymbol{\Sigma}) \| \mathcal{N}(\mathbf{0}, \mathbf{I})], \tag{7}$$

where $q_\phi(\mathbf{z}_l|\boldsymbol{x})$ is modeled as a Gaussian distribution with mean $\boldsymbol{\mu}$ and diagonal covariance $\boldsymbol{\Sigma}$, both of which are the outputs of the *Learngene*.

To ensure that the *Learngene* network exhibits generalization and that its output latent representations $\mathbf{z}_l$ possess class-invariant properties, we design an adversarial classifier (parameterized by $\vartheta$) on *Learngene* for adversarial optimization training:

$$\mathcal{L}_{adv} = -\mathbb{E}_{\mathbf{z}_l \sim q_\phi(\mathbf{z}_l|\boldsymbol{x})} \log q_\vartheta(y|\mathbf{z}_l), \tag{8}$$

where $q_\vartheta(y|\mathbf{z}_l)$ represents the softmax probability output by the adversarial classifier. To equip *Learngene* with strong generalization ability, our objective is to ensure that its latent representation $\mathbf{z}_l$ remains client-agnostic and unbiased with respect to class distributions. The key intuition is that, under label distribution skew heterogeneity, each client observes only a subset of classes with highly imbalanced frequencies. We explicitly regularize $\mathbf{z}_l$ by enforcing the output of an auxiliary adversarial classifier to follow a uniform label distribution. This uniformity constraint compels the module to discard class-dominant patterns unique to individual clients and instead retain only the invariant, globally shared knowledge of the data. Formally, the uniform adversarial objective is defined as:

$$\mathcal{L}_{adv}^u = -\mathbb{E}_{q_\phi(\mathbf{z}_l|\boldsymbol{x})} \left[ \frac{1}{K} \sum_{k=1}^{K} \log q_\vartheta(k|\mathbf{z}_l) \right]. \tag{9}$$

This strategy effectively avoids biased learning of specific categories within a single client, and can enhance the generalization training of the *Learngene* module to achieve cross-client collaborative learning. In summary, *Learngene* captures generalized invariant representations to achieve consistent optimization across clients, with the loss defined as: $\mathcal{L}_{GL} = \mathcal{L}_{kl} + \mathcal{L}_{adv} + \mathcal{L}_{adv}^u$.

### 4.4 ROBUST REPRESENTATION LEARNING VIA NOISY RECONSTRUCTION

The latent representations produced by *PersonaNet* and *Learngene*, denoted as $\mathbf{z}_p$ and $\mathbf{z}_l$, are first concatenated and then fed into the decoder $p_\theta(\boldsymbol{x}'|\mathbf{z})$. The decoder parameters $\theta$ are optimized by minimizing the reconstruction loss: $\mathcal{L}_{rec} = \|\boldsymbol{x} - \boldsymbol{x}'\|_2^2$. Although minimizing $\mathcal{L}_{rec}$ ensures that the generated sample $\boldsymbol{x}'$ closely approximates the original input $\boldsymbol{x}$, such high-fidelity reconstructions usually lack diversity. This may cause the learned *Learngene* module to overfit to specific data instances, thereby weakening its generalization ability. To alleviate this, we inject Gaussian noise into the reconstructed samples during classifier training to promote more robust and diverse gradient propagation during backpropagation: $\boldsymbol{x}_p = \boldsymbol{x}' + n$, where $n \sim \mathcal{N}(0, \sigma^2\mathbf{I})$. This perturbation can reduce the risk of reconstructing the original data by encouraging *Learngene* to capture more transferable and generalized representations. Subsequently, a local classifier $f_\omega(\cdot)$ is trained on both the original and augmented data to simulate the label prediction process. The overall classification loss is defined as:

$$\begin{aligned}
\mathcal{L}_{ce} &= \mathbb{E}_{(\boldsymbol{x},y)\sim\mathcal{D}_m} \ell\left(f_\omega\left(\boldsymbol{x}\right), y\right) \\
&+ \mathbb{E}_{(\boldsymbol{x}_p,y)\sim P(\boldsymbol{x}_p,y)} \ell\left(f_\omega\left(\boldsymbol{x}_p\right), y\right),
\end{aligned} \tag{10}$$

where $\mathcal{D}_m$ is the local data distribution for client $m$, $P(\boldsymbol{x}_p, y)$ represents the distribution of the perturbed data and labels, and $\ell(\cdot, \cdot)$ denotes the standard cross-entropy loss function. A detailed theoretical analysis is presented in Appendix A.2, with corresponding proofs in Appendix A.3, and the discussion and limitations of the proposed method are provided in Appendix A.4.

## 5 EXPERIMENTS

### 5.1 EXPERIMENT SETUP

**Dataset and data partition.** We conduct experiments on three standard FL benchmarks: SVHN (Netzer et al., 2011), CIFAR-10, and CIFAR-100 (Krizhevsky et al., 2009). To simulate realistic federated scenarios, we adopt two types of non-IID data partitions. For Dirichlet-based partitioning, training and test data are distributed across clients following a Dirichlet distribution with $\beta \in \{0.1, 0.4\}$ (Chen et al., 2021; Dai et al., 2022), inducing varying degrees of label skew. For shard-based partitioning, data is split by class into shards and unevenly assigned to clients, controlling heterogeneity via the number of classes per client. Specifically, we set $s \in \{4, 5\}$ for SVHN and CIFAR-10, and $s \in \{20, 30\}$ for CIFAR-100.

**Evaluation metrics.** We report the mean test accuracy of personalized models for all clients. The evaluation is based on two primary metrics: Local-T (i.e., using the local test data corresponding to each client's class distribution) and Global-T (i.e., using the union of all clients' local test data). These metrics are used to assess the model's personalization performance and generalization ability.

**Baselines.** We selected a series of state-of-the-art federated learning algorithms for comparison, including *Local*, which performs training locally without collaboration, and CFL methods designed to mitigate data heterogeneity, such as FedRep (Collins et al., 2021), FedNova (Wang et al., 2020),

FedBN (Li et al., 2021), and FedFed (Yang et al., 2024). Furthermore, DFL approaches, including DFedPGP (Liu et al., 2024), Fedcvae, and DisPFL (Dai et al., 2022), were used as baselines. All methods use the ResNet18 network as the backbone classifier. Detailed description is given in Appendix B.1.3.

## 5.2 EVALUATION RESULTS

**DRDFL is parameter-efficient.** Table 1 reports the parameters of the personalized module trained on each client and the communication cost per round transmitted to the server under different settings. In terms of communication efficiency, DRDFL significantly outperforms most state-of-the-art CFL and DFL methods across different settings. Classical CFL methods, such as FedRep, FedNova, and FedBN, require approximately 213 M parameters to be sent to the server for aggregation. In contrast, DRDFL only exchanges 0.58 M parameters, including the consensus lightweight *Learngene* module

Table 1: Comparison of parameter efficiency across different FL methods.

| Method | Personalized Params | # Comm. Params |
|--------|--------------------|----------------|
| *Local* | Full model | 0 M |
| FedRep | Output layer | 213.36 M |
| FedNova | Full model | 213.46 M |
| FedBN | BatchNorm layers | 213.06 M |
| FedFed | Full model | 454.58 M |
| DFedPGP | Output layer | 213.36 M |
| FedCVAE | Full model | 63.44 M |
| DisPFL | Masked model | 106.65 M |
| **DRDFL** | *PersonaNet* | **0.58 M** |

and a small set of global latent Gaussian representations. DFL-based approaches like DisPFL still rely on exchanging masked model components or low-level parameter updates, which remain considerably more costly than DRDFL. These comparisons highlight DRDFL's superior parameter efficiency and suitability for resource-constrained decentralized FL scenarios.

**DRDFL achieves generalization comparable to CFL and personalization competitive with DFL methods.** Tables 2 and 3 show that, compared to DFL methods on the same architecture, DRDFL delivers competitive personalization results with significantly fewer communication parameters. In the Dirichlet-based $\beta = 0.1$ setting, DRDFL outperforms the state-of-the-art DFedPGP method (+ 1.24%, 5.29%, 2.64% on SVHN, CIFAR-10, and CIFAR-100). DRDFL also achieves comparable generalization performance to server-based CFL methods on CIFAR-10, providing competitive results. While it performs slightly worse than the FedBN method (− 1.74%, 0.17% on CIFAR-100 dataset with $\beta = 0.1$, $s = 20$), which aggregates batch normalization models. However, DRDFL achieves the best performance that is higher than FedFed (+1.28%) on CIFAR-100 with $s = 30$. These results highlight the effectiveness of *Learngene* as a shared module for iterative optimization across clients, enabling the learning of generalized consensus knowledge in RDFL scenarios.

Table 2: Averaged test accuracy (%± std) across all clients' models under the Dirichlet-based non-IID setting. Note that **Bold** / Underline highlight the best / second-best approach.

| | SVHN | | | | CIFAR-10 | | | | CIFAR-100 | | | |
|--------|------|------|------|------|----------|------|------|------|-----------|------|------|------|
| | $\beta = 0.1$ | | $\beta = 0.4$ | | $\beta = 0.1$ | | $\beta = 0.4$ | | $\beta = 0.1$ | | $\beta = 0.4$ | |
| Method | Local-T | Global-T | Local-T | Global-T | Local-T | Global-T | Local-T | Global-T | Local-T | Global-T | Local-T | Global-T |
| *Local* | 94.68±0.2 | 26.55±0.3 | 94.37±0.3 | 56.69±0.4 | 89.12±0.3 | 23.97±0.2 | 76.12±0.4 | 36.81±0.3 | 63.93±0.2 | 11.19±0.2 | 46.44±0.3 | 16.07±0.2 |
| FedRep | 91.58±0.3 | 27.32±0.2 | 94.65±0.3 | 56.92±0.3 | 86.26±0.3 | 21.64±0.3 | 77.35±0.3 | 38.88±0.2 | 65.92±0.3 | 11.70±0.2 | 43.36±0.3 | 17.44±0.3 |
| FedNova | 95.65±0.2 | 31.65±0.3 | 93.66±0.3 | 56.69±0.3 | 90.80±0.3 | 25.56±0.2 | 79.89±0.3 | 39.93±0.2 | 65.94±0.3 | 11.12±0.2 | 45.68±0.2 | 16.06±0.3 |
| FedBN | 93.49±0.3 | 32.93±0.2 | 94.27±0.2 | **58.79**±0.3 | 86.38±0.3 | 25.01±0.3 | 80.90±0.3 | 39.87±0.2 | 63.70±0.3 | **14.96**±0.3 | 35.96±0.3 | **19.04**±0.3 |
| FedFed | 95.35±0.2 | 32.32±0.3 | 93.60±0.3 | 56.63±0.2 | 91.25±0.3 | 27.92±0.2 | 82.65±0.3 | 45.64±0.2 | 68.14±0.2 | 12.46±0.3 | 48.62±0.3 | 16.49±0.3 |
| DFedPGP | 95.93±0.3 | 30.17±0.2 | 92.57±0.3 | 54.60±0.3 | 87.57±0.2 | 25.41±0.3 | 78.01±0.3 | 42.67±0.2 | 70.20±0.3 | 11.23±0.2 | 43.10±0.2 | 16.72±0.2 |
| Fedcvae | 76.76±0.3 | 14.07±0.2 | 78.77±0.2 | 41.42±0.3 | 78.12±0.3 | 14.57±0.2 | 78.77±0.3 | 41.38±0.2 | 58.08±0.3 | 8.07±0.2 | 40.04±0.2 | 11.48±0.3 |
| DisPFL | 95.69±0.2 | 28.96±0.3 | 93.14±0.3 | 50.60±0.3 | 89.38±0.3 | 25.31±0.3 | 79.49±0.3 | 38.78±0.2 | 58.34±0.3 | 9.90±0.2 | 47.51±0.3 | 15.54±0.2 |
| DRDFL | **97.17**±0.2 | **33.04**±0.3 | **94.68**±0.3 | 57.87±0.2 | **92.86**±0.2 | **28.14**±0.2 | **85.93**±0.3 | **47.01**±0.3 | **72.84**±0.2 | 13.22±0.2 | **49.10**±0.3 | 17.55±0.2 |

Table 3: Averaged test accuracy across all clients' models under the Shard-based non-IID setting.

| | SVHN | | | | CIFAR-10 | | | | CIFAR-100 | | | |
|--------|------|------|------|------|----------|------|------|------|-----------|------|------|------|
| | $s = 4$ | | $s = 5$ | | $s = 4$ | | $s = 5$ | | $s = 20$ | | $s = 30$ | |
| Method | Local-T | Global-T | Local-T | Global-T | Local-T | Global-T | Local-T | Global-T | Local-T | Global-T | Local-T | Global-T |
| *Local* | 92.01±0.3 | 36.32±0.2 | 91.17±0.2 | 44.41±0.3 | 84.61±0.3 | 31.51±0.3 | 75.83±0.2 | 37.13±0.2 | 55.33±0.3 | 10.19±0.2 | 46.95±0.2 | 12.73±0.3 |
| FedRep | 93.62±0.2 | 36.55±0.3 | 94.49±0.2 | 46.36±0.3 | 88.80±0.3 | 33.84±0.2 | 82.18±0.3 | 38.02±0.3 | 56.55±0.2 | 10.40±0.2 | 52.41±0.3 | 14.26±0.2 |
| FedNova | 94.50±0.3 | 37.80±0.2 | 95.25±0.3 | 46.55±0.2 | 88.07±0.2 | 33.46±0.3 | 82.70±0.2 | 39.52±0.3 | 57.57±0.3 | 11.07±0.2 | 54.24±0.2 | 13.79±0.2 |
| FedBN | 92.93±0.2 | 38.93±0.3 | 94.28±0.3 | **48.34**±0.2 | 90.46±0.3 | 34.36±0.2 | 83.53±0.3 | 42.75±0.2 | 59.94±0.2 | **13.57**±0.3 | 55.83±0.2 | 14.82±0.2 |
| FedFed | 96.41±0.3 | 38.49±0.2 | 95.16±0.3 | 46.75±0.2 | 89.27±0.3 | 35.58±0.3 | 86.34±0.3 | 42.94±0.2 | 67.63±0.3 | 12.67±0.2 | 53.76±0.2 | 15.23±0.3 |
| DFedPGP | 91.89±0.2 | 37.01±0.3 | 92.31±0.3 | 45.74±0.2 | 87.06±0.3 | 32.15±0.3 | 80.49±0.2 | 38.59±0.2 | 69.33±0.2 | 13.35±0.3 | 58.25±0.3 | 13.63±0.2 |
| Fedcvae | 86.16±0.3 | 34.04±0.2 | 78.17±0.3 | 39.28±0.2 | 70.81±0.2 | 26.93±0.3 | 74.98±0.3 | 37.06±0.2 | 63.76±0.3 | 11.93±0.2 | 52.55±0.2 | 14.85±0.2 |
| DisPFL | 94.31±0.3 | 37.57±0.2 | 95.24±0.3 | 46.71±0.3 | 87.00±0.2 | 33.65±0.2 | 80.90±0.3 | 39.55±0.2 | 60.11±0.3 | 10.84±0.2 | 53.97±0.3 | 13.50±0.2 |
| DRDFL | **96.67**±0.2 | **39.93**±0.3 | **95.31**±0.3 | 47.69±0.2 | **92.25**±0.2 | **36.67**±0.3 | **89.52**±0.3 | **44.61**±0.3 | **71.19**±0.3 | 13.40±0.2 | **58.55**±0.2 | **16.51**±0.3 |

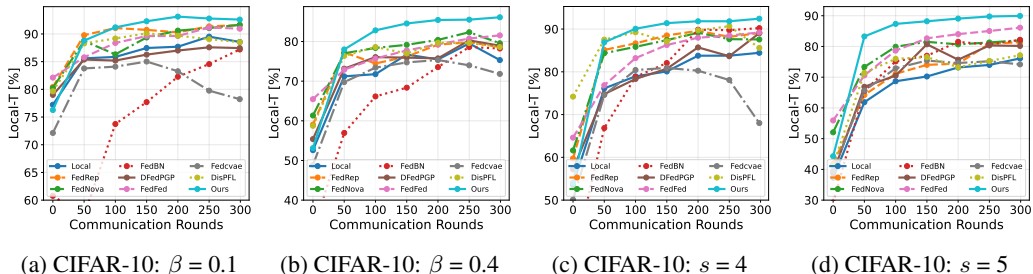

(a) CIFAR-10: $\beta = 0.1$     (b) CIFAR-10: $\beta = 0.4$     (c) CIFAR-10: $s = 4$     (d) CIFAR-10: $s = 5$

Figure 3: Comparison of Local-T curves for different methods under various non-IID partition settings on CIFAR-10 dataset.

**Convergence analysis.** We demonstrate the personalized performance of the model from a convergence perspective in Figure 3, which shows the performance curves of the FL method across different partitioning schemes on the CIFAR-10 dataset over communication rounds. Compared to other advanced methods, DRDFL achieves the best convergence speed under different partition settings and converges to higher personalized performance without introducing convergence-related problems. In particular, it is more significant on shard-based non-IID data partitions. The personalized performance of DRDFL is already higher than other methods at **Round 50** and gradually increases in the subsequent training stages to reach a convergence state.

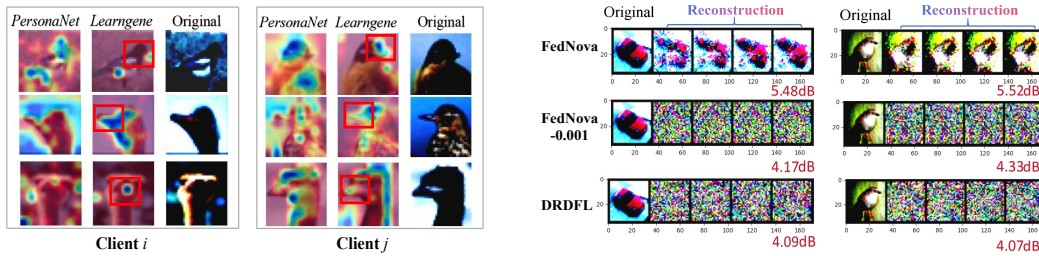

Figure 4: Visualization of **bird** class samples from CIFAR-10 across clients, with red boxes marking shared attention regions.

Figure 5: Reconstructed images from leaked information and corresponding PSNR (Hore & Ziou, 2010) values after 200 iterations.

**Grad-CAM (Selvaraju et al., 2017) visualization of *Learngene* and *PersonaNet* representations.** To further validate the DRDFL's capability in capturing both generalized and personalized information, we conduct Grad-CAM visualizations of the *PersonaNet* and *Learngene* modules on "bir" category image samples from different clients, as illustrated in Figure 4. The activation maps generated by *PersonaNet* reflect client-specific attention regions, highlighting personalized patterns learned by each client model. In contrast, the *Learngene* module consistently focuses on semantically meaningful and discriminative regions between clients, such as the head and beak of the bird. This observation confirms that *Learngene* is capable of learning generalized representations that maintain consistent focus on class-relevant semantic regions, regardless of the client-specific distribution variations.

**Robustness to gradient-based attack.** A recent approach called Deep Leakage from Gradients (DLG) (Zhu et al., 2019) of raises a crucial threat to the FL framework that aggregates the local gradients at the central server, DLG optimizes a dummy input to mimic shared local gradients, gradually approaching the original input sample, and repeatedly rehearses loss and gradient computations for data reconstruction. Instead, DRDFL transmit the *Learngene* and class Gaussian distributions that summarize client-relevant characteristics without retaining any recoverable instance-specific detail, making loss rehearsal infeasible for the attacker. We conduct reconstruction experiments using CIFAR-10 and CIFAR-100, comparing FedNova, FedNova with Gaussian noise, and DRDFL. Each attack is performed for 200 optimization steps. As illustrated in Figure 5, FedNova yields visually recognizable reconstructions, whereas DRDFL produces indistinguishable outputs with substantially lower PSNR values. Additional experiments for DRDFL are provided in Appendix B.2, including: computation overhead of the client, ablation studies of DRDFL components, applicability

to large-scale client populations, scalability to newly joined clients, and adaptability across various communication topologies.

## 6 CONCLUSIONS

In this paper, we propose a divide-and-conquer collaboration ring-topology Decentralized federated learning framework, which decouples the goals of generalization and personalization by designing two learning modules, *Learngene* and *PersonaNet*. The former uses adversarial learning to extract invariant representation, while the latter leverages Gaussian mixture learning to enhance class separability, achieving a dual benefit of both generalization and personalization. Extensive evaluations across multiple datasets validate the proposed method's effectiveness in achieving generalized and personalized performance under decentralized settings.

## ETHICS STATEMENT

This work builds upon publicly available benchmark datasets such as CIFAR-10 and CIFAR-100, which do not contain any personally identifiable or sensitive information. Our design does not introduce additional privacy concerns beyond existing FL frameworks, and we believe this work raises no direct ethical issues.

## REPRODUCIBILITY STATEMENT

We have taken substantial steps to ensure the reproducibility of our results. The details of the DRDFL framework, including model architectures, training procedures, hyperparameter configurations, dataset partition strategies, and evaluation metrics, are fully described in Appendix B.1. Additional ablation studies, scalability tests, and results under various communication topologies are also provided in Appendix B.2. To further facilitate replication, we will release the complete source code and scripts upon publication.

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

# A APPENDIX

## A.1 THE ELBO OF THE LOG-LIKELIHOOD OBJECTIVE

First, the *PersonaNet* network outputs a class representation $\mathbf{z}_p \sim p(\mathbf{z}_p, k)$, and the *Learngene* outputs a cross-class independent representation $\mathbf{z}_l \sim p(\mathbf{z}_l)$. Then, the decoder $p_\theta(x|\mathbf{z}_p, \mathbf{z}_l)$ takes the combination of $\mathbf{z}_p$ and $\mathbf{z}_l$ as input and maps the latent representations to images. Therefore, we decompose the joint distribution $p(x, \mathbf{z}_p, \mathbf{z}_l)$ as follows:

$$p(\boldsymbol{x}, \mathbf{z}_p, \mathbf{z}_l) = \sum_k p_\theta(\boldsymbol{x}|\mathbf{z}_p, \mathbf{z}_l) \, p(\mathbf{z}_p, k) \, p(\mathbf{z}_l) \tag{11}$$

By using Jensens inequality, the log -likelihood $\log p(\boldsymbol{x})$ can be written as:

$$
\begin{aligned}
\log p(\boldsymbol{x}) &= \log \iint p(\boldsymbol{x}, \mathbf{z}_p, \mathbf{z}_l) \, d\mathbf{z}_p d\mathbf{z}_l \\
&= \log \iint \sum_k p_\theta(\boldsymbol{x}|\mathbf{z}_p, \mathbf{z}_l) \, p(\mathbf{z}_p, k) \, p(\mathbf{z}_l) \, d\mathbf{z}_p d\mathbf{z}_l \\
&= \log \mathbb{E}_{q_\psi(\mathbf{z}_p, k|\boldsymbol{x}), q_\phi(\mathbf{z}_l|\boldsymbol{x})} \frac{p_\theta(\boldsymbol{x}|\mathbf{z}_p, \mathbf{z}_l) \, p(\mathbf{z}_p, k) \, p(\mathbf{z}_l)}{q_\psi(\mathbf{z}_p, k|\boldsymbol{x}) \, q_\phi(\mathbf{z}_l|\boldsymbol{x})} \\
&\geq \mathbb{E}_{q_\psi(\mathbf{z}_p, k|\boldsymbol{x}), q_\phi(\mathbf{z}_l|\boldsymbol{x})} \left[ \log \frac{p_\theta(\boldsymbol{x}|\mathbf{z}_p, \mathbf{z}_l) \, p(\mathbf{z}_p, k) \, p(\mathbf{z}_l)}{q_\psi(\mathbf{z}_p, k|\boldsymbol{x}) \, q_\phi(\mathbf{z}_l|\boldsymbol{x})} \right] \\
&= \mathbb{E}_{q_\psi(\mathbf{z}_p, k|\boldsymbol{x}), q_\phi(\mathbf{z}_l|\boldsymbol{x})} \left[ \log p_\theta(\boldsymbol{x}|\mathbf{z}_p, \mathbf{z}_l) \right] \\
&\quad + \mathbb{E}_{q_\psi(\mathbf{z}_p, k|\boldsymbol{x}), q_\phi(\mathbf{z}_l|\boldsymbol{x})} \left[ \log \frac{p(\mathbf{z}_p, k)}{q_\psi(\mathbf{z}_p, k|\boldsymbol{x})} \right] \\
&\quad + \mathbb{E}_{q_\psi(\mathbf{z}_p, k|\boldsymbol{x}), q_\phi(\mathbf{z}_l|\boldsymbol{x})} \left[ \log \frac{p(\mathbf{z}_l)}{q_\phi(\mathbf{z}_l|\boldsymbol{x})} \right] \\
&= \mathbb{E}_{q_\psi(\mathbf{z}_p, k|\boldsymbol{x}), q_\phi(\mathbf{z}_l|\boldsymbol{x})} \left[ \log p_\theta(\boldsymbol{x}|\mathbf{z}_p, \mathbf{z}_l) \right] \\
&\quad - D_{\mathrm{KL}}\left( q_\psi(\mathbf{z}_p, k|\boldsymbol{x}) \, \| \, p(\mathbf{z}_p, k) \right) \\
&\quad - D_{\mathrm{KL}}\left( q_\phi(\mathbf{z}_l|\boldsymbol{x}) \, \| \, p(\mathbf{z}_l) \right)
\end{aligned}
\tag{12}
$$

## A.2 THEORETICAL ANALYSIS

Before analyzing the convergence of DRDFL, we first introduce additional notation. Let $t$ denote the communication round among clients, and let $e \in \{0, 1, \ldots, E\}$ represent the local training epoch or iteration within each client. The iteration index $tE + e$ corresponds to the $e$-th local update in the $(t+1)$-th communication round. Specifically, $tE + 0$ refers to the point at which, in the $(t+1)$-th round, clients receive the *Learngene* in the $t$-th round prior to commencing local training. Conversely, $tE + E$ denotes the final iteration of local training, marking the completion of local updates in the $(t+1)$-th round. For simplicity, we assume that all models adopt a uniform learning rate $\eta$.

**Assumption 1. Lipschitz Smoothness.** Gradients of $m$-th client's local complete model $\boldsymbol{w}_m$ are L1-Lipschitz smooth (Tan et al., 2022; Yi et al., 2024),

$$
\begin{aligned}
\|\nabla \mathcal{L}_m^{t_1}(\boldsymbol{w}_m^{t_1}; \boldsymbol{x}, y) - \nabla \mathcal{L}_m^{t_2}(\boldsymbol{w}_m^{t_2}; \boldsymbol{x}, y)\| &\leq L_1 \|\boldsymbol{w}_m^{t_1} - \boldsymbol{w}_m^{t_2}\|, \\
\forall t_1, t_2 > 0, m \in \{0, 1, \ldots, M-1\}, (\boldsymbol{x}, y) &\in D_m.
\end{aligned}
\tag{13}
$$

The above formulation can be further expressed as:

$$\mathcal{L}_m^{t_1} - \mathcal{L}_m^{t_2} \leq \left\langle \nabla \mathcal{L}_m^{t_2}, \left(\boldsymbol{w}_m^{t_1} - \boldsymbol{w}_m^{t_2}\right) \right\rangle + \frac{L_1}{2} \left\| \boldsymbol{w}_m^{t_1} - \boldsymbol{w}_m^{t_2} \right\|_2^2. \tag{14}$$

**Assumption 2. Unbiased Gradient and Bounded Variance.** The client $m$'s random gradient $g_{\boldsymbol{w},m}^t = \nabla \mathcal{L}_m^t(\boldsymbol{w}_m^t; \xi_m^t)$ ($\xi$ is a batch of local data) is unbiased,

$$\mathbb{E}_{\xi_m^t \subseteq D_m}\left[ g_{\boldsymbol{w},m}^t \right] = \nabla \mathcal{L}_m^t(\boldsymbol{w}_m^t), \tag{15}$$

and the variance of $g_{\boldsymbol{w},m}^t$ is bounded by:

$$\mathbb{E}_{\xi_m^t \subseteq D_m} \left[ \|\nabla\mathcal{L}_m^t\left(\boldsymbol{w}_m^t;\xi_m^t\right) - \nabla\mathcal{L}_m^t\left(\boldsymbol{w}_m^t\right)\|_2^2 \right] \leq \sigma^2. \tag{16}$$

**Assumption 3. Bounded Parameter Variation from Ring-wise Propagation.** The parameter variations of the homogeneous *Learngene* $\phi_m^t$ and $\tilde{\phi}^t$ before and after receiving neighbor's is bounded as:

$$\left\|\tilde{\phi}^t - \phi_m^t\right\|_2^2 \leq \delta^2. \tag{17}$$

Based on the above assumptions, we can derive the following Lemma and Theorem.

**Lemma 1. Local Training.** Based on Assumptions 1 and 2, during $\{0,1,\ldots,E\}$ local iterations of the $(t+1)$-th FL training round, the loss of an arbitrary client's local model is bounded by:

$$\mathrm{E}\left[\mathcal{L}_{(t+1)E}\right] \leq \mathcal{L}_{tE+0} + \left(\frac{L_1\eta^2}{2} - \eta\right)\sum_{e=0}^{E-1}\|\nabla\mathcal{L}_{tE+e}\|_2^2 + \frac{L_1\eta^2\sigma^2}{2}. \tag{18}$$

**Lemma 2. Loss Bound after Receiving *Learngene*.** Given Assumptions 2 and 3, after the $(t+1)$-th local training round, the client's loss before and after receiving the lightweight *Learngene* from its neighbor is bounded by

$$\mathbb{E}\left[\mathcal{L}_{(t+1)E+0}\right] \leq \mathbb{E}\left[\mathcal{L}_{tE+1}\right] + \eta\delta^2. \tag{19}$$

**Theorem 1. One Communication Round of FL.** Based on Lemma 1 and Lemma 2, we get

$$\mathbb{E}\left[\mathcal{L}_{(t+1)E+0}\right] \leq \mathcal{L}_{tE+0} + \left(\frac{L_1\eta^2}{2} - \eta\right)\sum_{e=0}^{E}\|\nabla\mathcal{L}_{tE+e}\|_2^2 + \frac{L_1E\eta^2\sigma^2}{2} + \eta\delta^2. \tag{20}$$

**Theorem 2. Non-convex Convergence Rate of DRDFL.** Based on Theorem 1, for any client and an arbitrary constant $\epsilon > 0$, the following holds true:

$$\frac{1}{T}\sum_{t=0}^{T-1}\sum_{e=0}^{E-1}\|\nabla\mathcal{L}_{tE+e}\|_2^2 \leq \frac{\frac{1}{T}\sum_{t=0}^{T-1}\left[\mathcal{L}_{tE+0} - \mathbb{E}\left[\mathcal{L}_{(t+1)E+0}\right]\right] + \frac{L_1E\eta^2\sigma^2}{2} + \eta\delta^2}{\eta - \frac{L_1\eta^2}{2}} < \epsilon,$$
$$\text{s.t. } \eta < \frac{2\left(\epsilon - \delta^2\right)}{L_1\left(\epsilon + E\sigma^2\right)}. \tag{21}$$

Therefore, we conclude that any client's local model can converge at a non-convex rate $\epsilon \sim \mathcal{O}\left(\frac{1}{T}\right)$ under DRDFL.

A.3   THEORETICAL PROOF

A.3.1   PROOF FOR LEMMA 1

An arbitrary client $m$'s local model $\boldsymbol{w}$ can be updated by $\boldsymbol{w}_{t+1} = \boldsymbol{w}_t - \eta g_{\boldsymbol{w}_t}$ in the $(t+1)$-th round, and following Assumption 1, we can obtain:

$$\mathcal{L}_{t+1} \leq \mathcal{L}_t + \langle\nabla\mathcal{L}_{tE+0}, (\boldsymbol{w}_{tE+1} - \boldsymbol{w}_{tE+0})\rangle + \frac{L_1}{2}\|\boldsymbol{w}_{tE+1} - \boldsymbol{w}_{tE+0}\|^2$$
$$= \mathcal{L}_{tE+0} - \eta\langle\nabla\mathcal{L}_{tE+0}, g_{\boldsymbol{w},tE+0}\rangle + \frac{L_1\eta^2}{2}\|g_{\boldsymbol{w},tE+0}\|^2. \tag{22}$$

Taking the expectation of both sides of the inequality concerning the random variable $\xi_{tE+0}$, we obtain:

$$\mathbb{E}\big[\mathcal{L}_{tE+1}\big] \leq \mathcal{L}_{tE+0} - \eta\mathbb{E}\big[\langle\nabla\mathcal{L}_{tE+0}, g_{\boldsymbol{w},tE+0}\rangle\big] + \frac{L_1\eta^2}{2}\mathbb{E}\big[\|g_{\boldsymbol{w},tE+0}\|_2^2\big]$$

$$\overset{(a)}{\leq} \mathcal{L}_{tE+0} - \eta\|\nabla\mathcal{L}_{tE+0}\|_2^2 + \frac{L_1\eta^2}{2}\mathbb{E}\big[\|g_{\boldsymbol{w},tE+0}\|_2^2\big]$$

$$\overset{(b)}{\leq} \mathcal{L}_{tE+0} - \eta\|\nabla\mathcal{L}_{tE+0}\|_2^2 + \frac{L_1\eta^2}{2}\big(\mathbb{E}\big[\|g_{\boldsymbol{w},tE+0}\|_2^2 + \mathrm{Var}(g_{\boldsymbol{w},tE+0})\big]\big)$$

$$\overset{(c)}{\leq} \mathcal{L}_{tE+0} - \eta\|\nabla\mathcal{L}_{tE+0}\|_2^2 + \frac{L_1\eta^2}{2}\big(\|\nabla\mathcal{L}_{tE+0}\|_2^2 + \mathrm{Var}(g_{\boldsymbol{w},tE+0})\big) \tag{23}$$

$$\overset{(d)}{\leq} \mathcal{L}_{tE+0} - \eta\|\nabla\mathcal{L}_{tE+0}\|_2^2 + \frac{L_1\eta^2}{2}\big(\|\nabla\mathcal{L}_{tE+0}\|_2^2 + \sigma^2\big)$$

$$= \mathcal{L}_{tE+0} + \left(\frac{L_1\eta^2}{2} - \eta\right)\|\nabla\mathcal{L}_{tE+0}\|_2^2 + \frac{L_1\eta^2\sigma^2}{2},$$

where (a), (c), (d) follow Assumption 2. (b) follows $\mathrm{Var}(x) = \mathbb{E}[x^2] - \langle\mathbb{E}[x]^2\rangle$.

Taking the expectation of both sides of the inequality for the model $\boldsymbol{w}$ over $E$ iterations, we obtain

$$\mathbb{E}\big[\mathcal{L}_{tE+1}\big] \leq \mathcal{L}_{tE+0} + \left(\frac{L_1\eta^2}{2} - \eta\right)\sum_{i=1}^{E}\|\nabla\mathcal{L}_{tE+e}\|_2^2 + \frac{L_1E\eta^2\sigma^2}{2}. \tag{24}$$

### A.3.2 PROOF FOR LEMMA 2

$$\mathcal{L}_{(t+1)E+0} = \mathcal{L}_{(t+1)E} + \mathcal{L}_{(t+1)E+0} - \mathcal{L}_{(t+1)E}$$

$$\overset{(a)}{\approx} \mathcal{L}_{(t+1)E} + \eta\|\phi_{(t+1)E+0} - \phi_{(t+1)E}\|_2^2 \tag{25}$$

$$\overset{(b)}{\leq} \mathcal{L}_{(t+1)E} + \eta\delta^2,$$

where (a): we can use the gradient of parameter variations to approximate the loss variations, i.e., $\Delta\mathcal{L} \approx \eta \cdot \|\Delta\phi\|_2^2$. (b) follows Assumption 3. Taking the expectation of both sides of the inequality to the random variable $\xi$, we obtain

$$\mathbb{E}\left[\mathcal{L}_{(t+1)E+0}\right] \leq \mathbb{E}\left[\mathcal{L}_{tE+1}\right] + \eta\delta^2. \tag{26}$$

### A.3.3 PROOF FOR THEOREM 1

Substituting Lemma 1 into the right side of Lemma 2's inequality, we obtain

$$\mathbb{E}[\mathcal{L}_{(t+1)E+0}] \leq \mathcal{L}_{tE+0} + (\frac{L_1\eta^2}{2} - \eta)\sum_{e=0}^{E}\|\nabla\mathcal{L}_{tE+e}\|_2^2 + \frac{L_1E\eta^2\sigma^2}{2} + \eta\delta^2. \tag{27}$$

### A.3.4 PROOF FOR THEOREM 2

Interchanging the left and right sides of Eq. 27, we obtain

$$\sum_{e=0}^{E}\|\nabla\mathcal{L}_{tE+e}\|_2^2 \leq \frac{\mathcal{L}_{tE+0} - \mathbb{E}[\mathcal{L}_{(t+1)E+0}] + \frac{L_1E\eta^2\sigma^2}{2} + \eta\delta^2}{\eta - \frac{L_1\eta^2}{2}}. \tag{28}$$

Taking expectation over rounds $t = [0, T-1]$:

$$\frac{1}{T}\sum_{t=0}^{T-1}\sum_{e=0}^{E-1}\|\nabla\mathcal{L}_{tE+e}\|_2^2 \leq \frac{\frac{1}{T}\sum_{t=0}^{T-1}[\mathcal{L}_{tE+0} - \mathbb{E}[\mathcal{L}_{(t+1)E+0}]] + \frac{L_1E\eta^2\sigma^2}{2} + \eta\delta^2}{\eta - \frac{L_1\eta^2}{2}}. \tag{29}$$

Let $\Delta = \mathcal{L}_{t=0} - \mathcal{L}^* > 0$, then $\sum_{t=0}^{T-1}[\mathcal{L}_{tE+0} - \mathbb{E}[\mathcal{L}_{(t+1)E+0}]] \leq \Delta$, we get

$$\frac{1}{T}\sum_{t=0}^{T-1}\sum_{e=0}^{E-1}\|\nabla\mathcal{L}_{tE+e}\|_2^2 \leq \frac{\frac{\Delta}{T} + L_1E\eta^2\sigma^2 + \eta\delta^2}{\eta - \frac{L_1\eta^2}{2}}. \tag{30}$$

If this converges to a constant $\epsilon$, i.e.,

$$\frac{\Delta}{\eta - \frac{L_1\eta^2}{2}} + \frac{L_1 E\eta^2\sigma^2}{2(\eta - \frac{L_1\eta^2}{2})} + \frac{\eta\delta^2}{\eta - \frac{L_1\eta^2}{2}} < \epsilon, \tag{31}$$

then

$$T > \frac{\Delta}{\epsilon\left(\eta - \frac{L_1\eta^2}{2}\right) - \frac{L_1 E\eta^2\sigma^2}{2} - \eta\delta^2}. \tag{32}$$

Since $T > 0$, $\Delta > 0$, we can get solving the above inequality yields:

$$\epsilon\left(\eta - \frac{L_1\eta^2}{2}\right) - \frac{L_1 E\eta^2\sigma^2}{2} - \eta\delta^2 > 0. \tag{33}$$

After solving the above inequality, we can obtain:

$$\eta < \frac{2(\epsilon - \delta^2)}{L_1(\epsilon + E\sigma^2)}. \tag{34}$$

Since $\epsilon$, $L_1$, $\sigma^2$, $\delta^2$ are all constants greater than $0$, $\eta$ has solutions. Therefore, when the learning rate $\eta$ satisfies the above condition, any client's local complete model can converge. Notice that the learning rate of the local complete model involves $\{\eta_\psi, \eta_\phi, \eta_\theta, \eta_\omega\}$, so it's crucial to set reasonable them to ensure model convergence. Since all terms on the right side of Eq. 30 except for $\Delta/T$ are constants, $\Delta$ is also a constant, DRDFL's non-convex convergence rate is $\epsilon \sim \mathcal{O}\left(\frac{1}{T}\right)$.

## A.4 DISCUSSION AND LIMITATIONS

Algorithm 1 outlines the optimization process of the $m$-th client's local model under the DRDFL framework with ring-topology decentralized training. The computational cost of DRDFL is primarily focused on local representation learning within each client, where both the *PersonaNet* and the *Learngene* are jointly optimized using client-specific data. Notably, each client performs a lightweight parameter exchange, limited to the *Learngene* module and class distribution statistics, with a single neighbor per communication round, avoiding the overhead of full model synchronization. This design ensures high scalability and efficiency, making the framework suited for large-scale federated systems with limited communication bandwidth. In real-world distributed systems, ring-based communication typically incorporates basic fault tolerance mechanisms during deployment. We can adopt a standard and simple solution based on a timeout fault detector. If a node does not respond within a threshold time, its upstream node will bypass that node and directly connect to its subsequent nodes, effectively ensuring uninterrupted system training. This improves system robustness without sacrificing communication efficiency.

---

**Algorithm 1:** DRDFL: Divide-and-conquer Collaboration for Ring-topology Decentralized Federated Learning

---

**Input:** Total number of devices $M$, total number of communication rounds $T$, local learning rate $\eta$, total number of classes $K$, client model $\boldsymbol{w}_m = [\psi_m, \phi_m, \theta_m, \omega_m]$, parameter $\alpha$.

**Output:** Updated *Learngene* $\phi_m$ and statistics $\{(\boldsymbol{\mu}_k^{(m)}, \boldsymbol{\Sigma}_k^{(m)})\}_{k=1}^K$ for each client.

1 **for** $t = 0$ *to* $T - 1$ **do**
2    **for** *each client* $m$ **do**
3      Let $n_m = (m - 1 + M) \mod M$ denote the previous neighbor in the ring.
4      **Receive:** $\tilde{\phi}$ and $\{(\tilde{\boldsymbol{\mu}}_k, \tilde{\boldsymbol{\Sigma}}_k)\}_{k=1}^K$ from neighbor $n_m$.
5      Update local statistics via EMA:
6      $\boldsymbol{\mu}_k^{(m)} \leftarrow \alpha \boldsymbol{\mu}_k^{(m)} + (1 - \alpha)\tilde{\boldsymbol{\mu}}_k$
7      $\boldsymbol{\Sigma}_k^{(m)} \leftarrow \alpha \boldsymbol{\Sigma}_k^{(m)} + (1 - \alpha)\tilde{\boldsymbol{\Sigma}}_k$
8      Set $\phi_m \leftarrow \tilde{\phi}$ and sample a batch of local data $\xi_m$.
9      *PersonaNet* **Execution:**
10      $\psi_m \leftarrow \psi_m - \eta \nabla_{\psi_m} \mathcal{L}_{PR}(\boldsymbol{\mu}_k^{(m)}, \boldsymbol{\Sigma}_k^{(m)}; \xi_m)$
11      $\boldsymbol{\mu}_k^{(m)}, \boldsymbol{\Sigma}_k^{(m)} \leftarrow -\nabla_{\boldsymbol{\mu}_k^{(m)}, \boldsymbol{\Sigma}_k^{(m)}} \mathcal{L}_{PR}$
12      $\mathbf{z}_p \leftarrow \psi_m(\xi_m)$
13      *Learngene* **Execution:**
14      $\phi_m \leftarrow \phi_m - \eta \nabla_{\phi_m} \mathcal{L}_{GL}(\xi_m)$
15      $\mathbf{z}_l \leftarrow \phi_m(\xi_m)$
16      *Decoder* **and** *Classifier* **Execution:**
17      $\xi'_m \leftarrow \theta_m(\mathbf{z}_p, \mathbf{z}_l)$
18      $\theta_m \leftarrow \theta_m - \eta \nabla_{\theta_m} \mathcal{L}_{rec}(\xi_m, \xi'_m)$
19      $\omega_m \leftarrow \omega_m - \eta \nabla_{\omega_m} \mathcal{L}_{ce}(\xi_m, \xi'_m)$
20      **Send:** updated $\tilde{\phi} \leftarrow \phi_m$ and $\{(\boldsymbol{\mu}_k^{(m)}, \boldsymbol{\Sigma}_k^{(m)})\}$ to next neighbor $(m + 1) \mod M$.
21    **end**
22 **end**

---

## B EXPERIMENTAL SUPPLEMENT

### B.1 EXPERIMENT SETUP

#### B.1.1 IMPLEMENTATION

We implemented the proposed method and the considered baselines in PyTorch. The models are trained using ResNet-18 in a simulated decentralized ring-topology federated learning environment with multiple participating clients. By default, the number of clients is set to 20, the learning rate is set to 1e-3, the number of global communication rounds is set to 300, the number of local update epochs is set to 5, and the batch size is set to 64. Both the centralized and decentralized federated learning methods required all 20 clients to participate in the training process for collaborative learning. Following (Chung, 1996; Grebenkov & Serror, 2014; Guo et al., 2024a), we set the parameter $\alpha$ in EMA to 0.99 to learn global class-related information. We set $\sigma^2 = 0.15$ based on validation analysis of noise hyperparameter experiments, as listed in Table 4. The main experimental setup involves 20 clients collaborating in training, while the ablation study extends the analysis to 50 clients. The centralized federated learning baseline methods are evaluated within a server-supported framework, whereas the decentralized federated learning baselines are implemented under a ring topology for subsequent experimental comparisons.

Table 4: Ablation study on the noise variance $\sigma^2$ in DRDFL.

| Noise Variance $\sigma^2$ | 1.0 | 1.5 | 2.0 | 3.0 |
|---|---|---|---|---|
| **Local-T (%)** | 92.21±0.2 | **92.89**±0.2 | 91.03±0.2 | 90.03±0.1 |

### B.1.2 DATASET AND DATA PARTITION

The SVHN dataset, designed for digit classification, contains 600,000 $32 \times 32$ RGB images of printed digits extracted from Street View house numbers. For our experiments, we utilize a subset comprising 33,402 images for training and 13,068 images for testing. CIFAR-10 is a comprehensive image dataset comprising 10 classes, with each class containing 6,000 samples of size $32 \times 32$. Similarly, CIFAR-100 is an extended version with 100 classes, where each class includes 600 samples of the same size, offering finer granularity for image classification tasks.

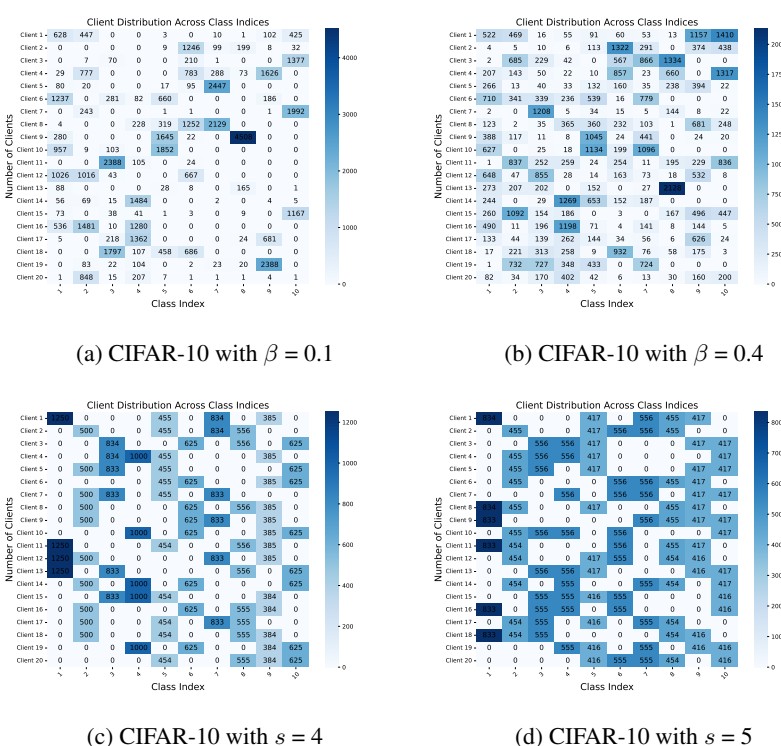

(a) CIFAR-10 with $\beta = 0.1$      (b) CIFAR-10 with $\beta = 0.4$

(c) CIFAR-10 with $s = 4$      (d) CIFAR-10 with $s = 5$

Figure 6: The non-IID data distribution simulated on different clients based on the CIFAR-10 dataset within the RDfl architecture.

Figure 6 illustrates the data distributions under different non-IID settings based on the CIFAR-10 dataset. Figures 6a and 6b show the client data distributions in the Dirichlet-based non-IID scenario, where both the class distribution and the number of samples vary across classes. In contrast, Figures 6c and 6d represent the Shard-based non-IID scenario, where each client has a distinct class distribution, but the number of samples per class remains identical. Both scenarios effectively simulate the problem of label distribution shift in data heterogeneity. Moreover, the test data shares the same class distribution as the training data but is composed of different samples, thereby modeling the feature distribution shift inherent in data heterogeneity.

### B.1.3 BASELINES

- **Local** is the direct solution to the personalized federated learning problem. Each client only performs SGD on their own data. For the sake of consistency, we take 5 epochs of local training as one communication round.

- **FedRep** (Collins et al., 2021) is a classic personalized federated learning method. It achieves personalized model training by sharing part of the model with the server during communication and training a personalized head locally. In our setup, the number of locally shared model training epochs is set to 4, and the number of personalization epochs is set to 1.

- **FedNova** (Wang et al., 2020) employs a normalized averaging approach to eliminate objective inconsistency while maintaining fast error convergence. This method ensures that models trained on non-IID data reduce objective inconsistencies, thereby improving the generalization performance of the global model.

- **FedBN** (Li et al., 2021) is a federated learning method based on the personalization of Batch Normalization (BN). Each client retains its personal BN layer statistics, including mean and variance, while other model parameters, such as weights and biases of convolutional and fully connected layers, are aggregated and shared among clients.

- **FedFed** (Yang et al., 2024) introduces a data-driven approach that divides the underlying data into performance-sensitive features (which contribute significantly to model performance) and performance-robust features (which have limited impact on model performance). Performance-sensitive features are globally shared to mitigate data heterogeneity, while performance-robust features are retained locally, facilitating personalized private models.

- **DFedPGP** (Liu et al., 2024) is a state-of-the-art personalized decentralized federated learning (DFL) method. It personalizes the linear classifier of modern deep models to tailor local solutions and learns consensus representations in a fully decentralized manner. Clients share gradients only with a subset of neighbors based on a directed and asymmetric topology, ensuring resource efficiency and enabling flexible choices for better convergence.

- **Fedcvae** is a comparative method we propose based on Conditional Variational Autoencoders (CVAE) (Sohn et al., 2015) and a decentralized federated learning architecture. Each client uses its private dataset trains a pretrained model $g_\varphi(\cdot)$ to obtain the prior distribution and the CVAE model $f_w(\cdot)$ and a classifier $C_\omega(\cdot)$ until convergence. The CVAE consists of an encoder $E_\phi(\cdot)$, a decoder $D_\theta(\cdot)$ with parameters denoted as $w = [\phi, \theta]$. Collaborative learning among clients is achieved by using the pretrained model as the shared interaction information. Previous research on CVAE has explored its application in defense against malicious clients (Wen et al., 2020; Gu & Yang, 2021). In one-shot federated learning (Heinbaugh et al., 2023), an ensemble dataset is constructed at the server to train a server-side classifier. In federated learning frameworks (Kasturi et al., 2022) based on VAE, client-generated data is aggregated at the server to train a global model. However, this approach is different from our learning goals and the decentralized learning scenario we are focusing on.

- **DisPFL** (Dai et al., 2022) is a classical personalized federated learning method in distributed scenarios. It uses personalized sparse masks to customize edge-local sparse models. During point-to-point communication, each local model maintains a fixed number of active parameters throughout the local training process, reducing communication costs.

### B.2 Additional Experimental Results

#### B.2.1 Convergence analysis

In Figure 7, we present a comparative analysis of personalized performance across various methods on the CIFAR-100 dataset under different non-IID settings. Similar to the trends observed in Figure 3 for the CIFAR-10 dataset, our method demonstrates a smooth convergence curve and outperforms other approaches in most cases. In the CIFAR-100 setting with $s = 20$, although the DFedPGP method achieves higher performance in some rounds, it exhibits more fluctuations. Particularly, in the CIFAR-100 setting with $\alpha = 0.1$ and $s = 30$, our method achieves higher accuracy with fewer communication rounds, highlighting its superior convergence speed. The results demonstrate that our proposed method, consistently outperforms other baseline approaches, such as FedRep, FedNova, FedBN, and DisPFL, in terms of Local-T, which indicates the model's ability to personalize effectively across clients. When considering convergence behavior, the proposed method also demonstrates faster convergence compared to the other methods. The model reaches higher Local-T with fewer communication rounds, highlighting its efficiency in both convergence speed and resource utilization.

#### B.2.2 Ablation study

In Table 5, we present the impact of different components on the overall method, evaluated using the Local-T and Global-T metrics for the CIFAR-10 dataset with different data partitions. When

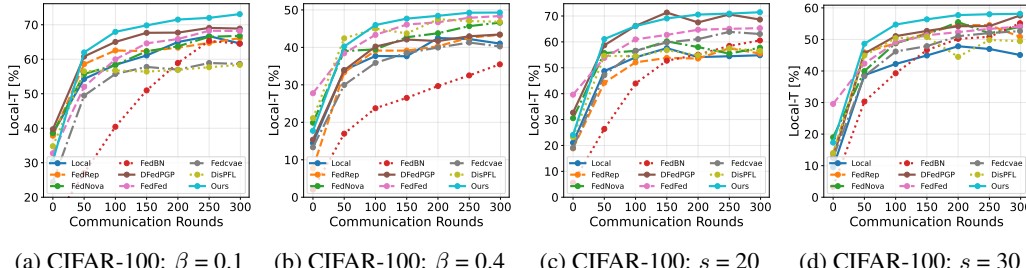

(a) CIFAR-100: $\beta = 0.1$    (b) CIFAR-100: $\beta = 0.4$    (c) CIFAR-100: $s = 20$    (d) CIFAR-100: $s = 30$

Figure 7: Comparison of Local-T curves for different methods under various non-IID partition settings on CIFAR-100 dataset.

Table 5: Ablation studies for DRDFL on CIFAR-10 dataset.

| Settings | $\mathcal{L}_{PR}$ | $\mathcal{L}_{GL}$ | $s = 4$ | | $\beta = 0.1$ | |
| | | | Local-T | Global-T | Local-T | Global-T |
|---|---|---|---|---|---|---|
| DRDFL w/o $\mathcal{L}_{PR}$ | ✗ | ✓ | 90.26±0.2 | 36.42±0.1 | 90.01±0.3 | 27.84±0.2 |
| DRDFL w/o $\mathcal{L}_{GL}$ | ✓ | ✗ | 91.39±0.1 | 34.32±0.3 | 92.63±0.2 | 26.84±0.2 |
| DRDFL | ✓ | ✓ | **92.25**±0.2 | **36.67**±0.3 | **92.86**±0.2 | **28.14**±0.2 |

the DRDFL method does not include the personalized $\mathcal{L}_{PR}$ component, its performance on Local-T is significantly worse, while Global-T remains roughly unchanged. In contrast, omitting the $\mathcal{L}_{GL}$ component, which controls for generalization invariant representations, slightly decreases Global-T performance.

### B.2.3 COMPUTATION OVERHEAD OF THE CLIENT

We provide two views to demonstrate the limited costs of extra computation, i.e., Training Time and FLOPs (Floating Point Operations). We empirically measure the training time of both the backbone classifier and the additional modules introduced by DRDFL. The classifier architecture is identical to that adopted in the baseline methods. As reported in Table 6, the additional modules incur less than 5% of the training time per batch (batch size = 32) compared to the classifier, indicating that the extra training cost is negligible. As shown in Table 7, the generator and its associated modules incur only 899.02 MFLOPs, whereas the classifier requires 17,872.58 MFLOPs. Thus, the additional FLOPs introduced by DRDFL constitute less than 5% of the overall computational cost.

Table 6: Training time per batch.

| Module | Time (s) |
|---|---|
| Classifier | 2.12 |
| Additional modules | 0.08 |

Table 7: FLOPs comparison.

| Module | MFLOPs |
|---|---|
| Classifier | 17,872.58 |
| Additional modules | 899.02 |

### B.2.4 APPLICABILITY TO LARGE-SCALE CLIENT POPULATIONS

To further validate the applicability of DRDFL across different client scales, we perform collaborative learning with 50 clients and compare it to the FedRep method in CFL and the DFedPGP method in DFL, as shown in Figure 8. DRDFL achieves significant improvements in both convergence speed and performance on the Local-T and Global-T metrics. The numbers in the figure represent the average values of the last 10 rounds, with DRDFL outperforming FedRep by 7.37% on Local-T and slightly outperforming it by 2.53% on Global-T.

### B.2.5 GRAD-CAM VISUALIZATION OF *Learngene* AND *PersonaNet* REPRESENTATIONS.

To further examine the distinct roles of *PersonaNet* and the *Learngene* module, we visualize their Grad-CAM activation maps on "cat" samples collected from randomly selected heterogeneous clients. As shown in Figure 9, the activation maps generated by *PersonaNet* primarily highlight

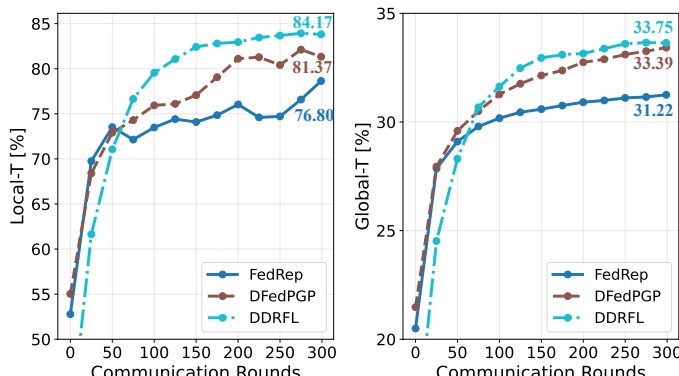

Figure 8: Comparison of Local-T and Global-T curves for different personalized methods on CIFAR-10 with $s = 4$ across 50 clients.

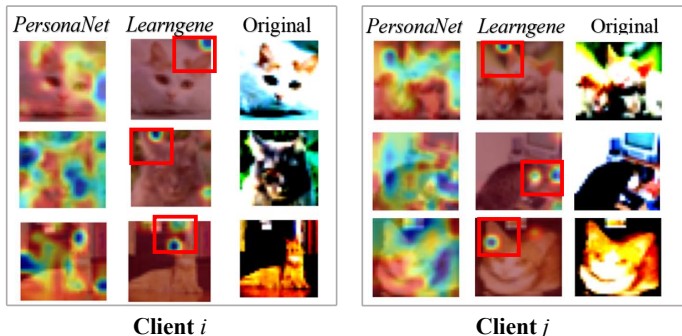

Figure 9: Visualization of **cat** class samples from CIFAR-10 across clients, with red boxes marking shared attention regions.

client-specific discriminative boundaries, reflecting the personalized decision-making focus of each client determined by its private data distribution. In contrast, the *Learngene* module consistently focuses on meaningful, representative, and distinguishable regions related to the category, such as the ears of a cat. This observation confirms that *Learngene* can capture consensus knowledge of class consistency, enabling it to maintain a stable focus on category-related structures while being unaffected by changes in client-specific distributions.

### B.2.6 SCALABILITY TO NEWLY JOINED CLIENTS

The ring topology offers excellent scalability, enabling new clients to dynamically join the federated learning system. However, this flexibility also introduces a new challenge: how to effectively initialize models for newly added clients. The *Learngene* module we designed, which encapsulates generalized knowledge-capturing transferable and generalizable representations can seamlessly adapt to unknown clients. The specific initialization process of the new client is shown in Algorithm 2.

---

**Algorithm 2:** New Client Initialization in DRDFL

---

**Input:** New client $m$, neighbor index $n_m$, received $\tilde{\phi}$, received global priors $\{(\tilde{\boldsymbol{\mu}}_k, \tilde{\boldsymbol{\Sigma}}_k)\}_{k=1}^{K}$
**Output:** Initialized model $\boldsymbol{w}_m = [\psi_m, \phi_m, \theta_m, \omega_m]$
1 Initialize $\phi_m \leftarrow \tilde{\phi}$
2 Initialize $\psi_m, \theta_m, \omega_m$ with random weights
3 Set $\boldsymbol{\mu}_k^{(m)} \leftarrow \tilde{\boldsymbol{\mu}}_k, \quad \boldsymbol{\Sigma}_k^{(m)} \leftarrow \tilde{\boldsymbol{\Sigma}}_k$
4 Train on local data $\xi_m$ using Algorithm 1 with fixed global priors for the first $T_{init} = 5$ rounds
5 **return** $\boldsymbol{w}_m$

---

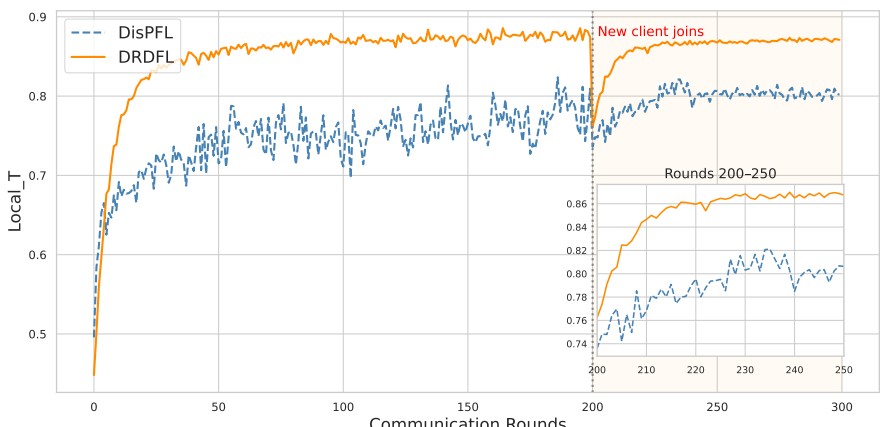

Figure 10: Visualization the average performance of new clients joining the ring-topology federated learning system on CIFAR-10. DRDFL provides strong initialization using the optimized *Learngene* and global priors, leading to rapid convergence.

We empirically validate this hypothesis through a two-stage experimental setup. In the first stage, a ring-topology federated learning system with 15 clients undergoes 200 rounds of collaborative training to ensure convergence. In the second stage, five new clients with previously unseen data distributions are introduced into the system. The average performance of two methods on participated clients is illustrated in Figure 10. It is clear that our proposed DRDFL method leverages the optimized *Learngene* and global Gaussian information to provide strong model initialization for the new clients, significantly accelerating their convergence. In contrast, DisPFL maintains a fixed number of active parameters and exhibits unstable performance when adapting to new clients during collaborative training.

### B.2.7 ADAPTABILITY ACROSS VARIOUS COMMUNICATION TOPOLOGIES

To evaluate the adaptability of our method under different communication topologies, we extend its core design to both the fully-connected topology (Figure 11 (a)) and a dynamically-varying connected topology (Figure 11 (b)). In the dynamically-varying connected topology, where each client is allowed to communicate only with a limited set of randomly selected neighbors that may differ across communication rounds. Specifically, each client averages the received updates from its connected peers via the *Learngene* module before performing local optimization on its private dataset. The corresponding experimental results are listed in Table 8.

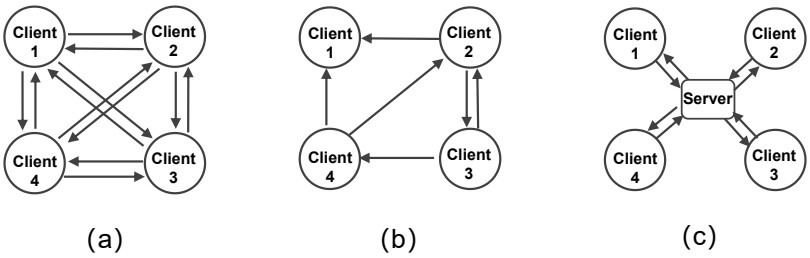

Figure 11: Illustrations of communication topologies. (a-b) correspond to decentralized settings, where (a) denotes the fully-connected topology and (b) the dynamically-varying topology. (c) depicts the centralized parameter-server architecture.

As shown in Table 8, the fully connected topology attains the highest Local-T and Global-T scores due to dense communication. By contrast, the ring topology achieves comparable performance within the same order of magnitude while requiring only linear communication overhead. From a practical standpoint, these results highlight a favorable efficiency–accuracy trade-off for ring-based

decentralized networks: they reduce redundant transmission complexity from quadratic $O(N^2)$ to linear $O(N)$ while maintaining nearly the same personalized and global performance as fully connected networks. This property makes ring topologies particularly attractive for large-scale deployments, such as vehicle-to-vehicle and IoT systems, where communication bandwidth and scalability are critical constraints.

Table 8: Results on CIFAR-100 under decentralized federated learning with various communication topologies.

| Method | $\beta = 0.1$ | | $s = 20$ | |
|---|---|---|---|---|
| | Local-T | Global-T | Local-T | Global-T |
| Ours-Ring Connected | 72.84±0.2 | 13.22±0.2 | 71.19±0.3 | 13.40±0.2 |
| Ours-dynamically Connected | 72.18±0.2 | 13.78±0.2 | 71.32±0.3 | 13.61±0.2 |
| Ours-Fully Connected | **74.02**±0.2 | **14.26**±0.3 | **71.89**±0.2 | **14.52**±0.2 |

Furthermore, to assess the scalability of our framework in the centralized topology (Figure 11 (c)), we extend DRDFL to a centralized federated learning setting (denoted as DRDFL-CFL) and report its performance in Table 9. DRDFL-CFL consistently achieves superior generalization performance (Global-T) compared to its ring-topology counterpart, demonstrating the benefit of more efficient global information exchange enabled by server coordination. Notably, DRDFL-CFL outperforms all other centralized baselines, including FedBN, FedNova, and FedFed, across both Dirichlet settings ($\beta = 0.1$ and $\beta = 0.4$).

Table 9: Comparison of Local-T and Global-T between CFL and DFL variants of DRDFL on CIFAR-10 under different Dirichlet non-IID settings.

| Setting | Method | $\beta = 0.1$ | | $\beta = 0.4$ | |
|---|---|---|---|---|---|
| | | Local-T | Global-T | Local-T | Global-T |
| CFL | **DRDFL-CFL** | 92.75±0.2 | **28.20**±0.2 | 85.21±0.3 | **47.61**±0.3 |
| DFL | **DRDFL** | **92.86**±0.2 | 28.14±0.2 | **85.93**±0.3 | 47.01±0.3 |

In contrast, the ring-based DRDFL achieves the highest personalization performance (Local-T), reflecting the advantage of preserving local adaptation in decentralized environments. This aligns with intuition: centralized aggregation can introduce global bias that compromises client-specific learning, while fully decentralized training better retains local characteristics. The proposed method supports scalable deployment across different communication topologies, including but not limited to RDFL. The key distinction between centralized FL (CFL) and decentralized FL (DFL) in our framework lies in the update strategy of the *Learngene* module and class Gaussian statistics (mean and variance): DFL employs exponential moving averages (EMA), whereas CFL adopts aggregation-based averaging, as shown in Algorithm 3. This flexibility enables DRDFL to generalize beyond ring topologies, establishing it as a robust and communication-efficient framework for addressing data heterogeneity across diverse federated learning settings.

In addition, we extend our decentralized federated learning framework to the classical large-scale CIFAR-100 dataset, considering both the fully connected topology, in which all nodes communicate with each other, and the partially connected topology, where each client can only communicate with a restricted set of randomly selected neighbors that may vary across communication rounds.

### B.2.8 CONVERGENCE ANALYSIS FOR RESPONSES TO Q2, 3, 5, AND 8

As shown in Figure 12, the proposed DRDFL consistently exhibits faster convergence and higher asymptotic accuracy than the baseline methods across all evaluated scenarios. Under partial participation (Q2) and extreme non-IID settings with no label overlap (Q3), DRDFL converges rapidly within the early communication rounds and maintains stable performance with limited fluctuations. On the more challenging TinyImageNet dataset with a ViT-B/16 backbone (Q5), DRDFL still achieves smoother convergence and a higher final accuracy than FedPGP. Compared with FedWSL (Fig. 12(d)), DRDFL reaches a stable performance significantly earlier and at a higher accuracy level.

---

**Algorithm 3:** Divide-and-conquer Collaboration for Centralized Federated Learning (CFL)

---

**Input:** Total number of devices $M$, total communication rounds $T$, local learning rate $\eta$, total number of classes $K$, local model $\boldsymbol{w}_m = [\psi_m, \phi_m, \theta_m, \omega_m]$, global *Learngene* $\tilde{\phi}$, global statistics $\{(\tilde{\boldsymbol{\mu}}_k, \tilde{\boldsymbol{\Sigma}}_k)\}_{k=1}^K$.

**Output:** Updated global *Learngene* $\tilde{\phi}$ and global statistics $\{(\tilde{\boldsymbol{\mu}}_k, \tilde{\boldsymbol{\Sigma}}_k)\}_{k=1}^K$.

---

1 **for** $t = 0$ *to* $T - 1$ **do**

2    **for** *each client $m$ in parallel* **do**

3      **Receive:** global *Learngene* $\tilde{\phi}$ and global statistics $\{(\tilde{\boldsymbol{\mu}}_k, \tilde{\boldsymbol{\Sigma}}_k)\}$.

4      Set $\phi_m \leftarrow \tilde{\phi}$, $\boldsymbol{\mu}_k^{(m)} \leftarrow \tilde{\boldsymbol{\mu}}_k$, $\boldsymbol{\Sigma}_k^{(m)} \leftarrow \tilde{\boldsymbol{\Sigma}}_k$.

5      Sample a batch of local data $\xi_m$.

6      *PersonaNet* **Execution:**

7      $\psi_m \leftarrow \psi_m - \eta \nabla_{\psi_m} \mathcal{L}_{PR}(\boldsymbol{\mu}_k^{(m)}, \boldsymbol{\Sigma}_k^{(m)}; \xi_m)$

8      $\boldsymbol{\mu}_k^{(m)}, \boldsymbol{\Sigma}_k^{(m)} \leftarrow -\nabla_{\boldsymbol{\mu}_k^{(m)}, \boldsymbol{\Sigma}_k^{(m)}} \mathcal{L}_{PR}$

9      $\mathbf{z}_p \leftarrow \psi_m(\xi_m)$

10      *Learngene* **Execution:**

11      $\phi_m \leftarrow \phi_m - \eta \nabla_{\phi_m} \mathcal{L}_{GL}(\xi_m)$

12      $\mathbf{z}_l \leftarrow \phi_m(\xi_m)$

13      *Decoder* **and** *Classifier* **Execution:**

14      $\xi_m' \leftarrow \theta_m(\mathbf{z}_p, \mathbf{z}_l)$

15      $\theta_m \leftarrow \theta_m - \eta \nabla_{\theta_m} \mathcal{L}_{rec}(\xi_m, \xi_m')$

16      $\omega_m \leftarrow \omega_m - \eta \nabla_{\omega_m} \mathcal{L}_{ce}(\xi_m, \xi_m')$

17    **end**

18    **Server Aggregation:**

19    $\tilde{\phi} \leftarrow \frac{1}{\sum_{m=1}^M |\mathcal{D}_m|} \sum_{m=1}^M |\mathcal{D}_m| \cdot \phi_m$

20    $\tilde{\boldsymbol{\mu}}_k \leftarrow \frac{1}{\sum_{m=1}^M |\mathcal{D}_m|} \sum_{m=1}^M |\mathcal{D}_m| \cdot \boldsymbol{\mu}_k^{(m)}$, $\tilde{\boldsymbol{\Sigma}}_k \leftarrow \frac{1}{\sum_{m=1}^M |\mathcal{D}_m|} \sum_{m=1}^M |\mathcal{D}_m| \cdot \boldsymbol{\Sigma}_k^{(m)}$

21 **end**

---

Figure 12: Convergence analysis for responses to Q2, 3, 5, and 8.

