# OpenReview forum: "DRDFL: Divide-and-conquer Collaboration for Efficient Ring-topology Decentralized Federated Learning"
_ICLR.cc/2026/Conference — ICLR 2026 Conference Withdrawn Submission_

### Official Review · Reviewer_LsAK · 2025-10-18

**Soundness:** 2
**Presentation:** 2
**Contribution:** 2
**Rating:** 4
**Confidence:** 3

**Summary:**

This paper proposes DRDFL, a divide-and-conquer framework for Ring-topology Decentralized Federated Learning. The method aims to tackle data heterogeneity and sparse communication challenges that arise in peer-to-peer FL systems without a central server. DRDFL introduces two modules: 1) a transferable Learngene module to encapsulate consensus knowledge addressing the label distribution skew. 2) a local PersonaNet module to mitigate feature distribution skew for local personalized feature modeling via Gaussian mixtures.

**Strengths:**

1) Decentralized federated learning is a timely and important research direction, as it mitigates several inherent limitations of centralized FL most notably the single point of failure and heavy reliance on a central coordinator. Addressing data heterogeneity among clients is also a key open challenge in this area, and this work takes a step toward that goal.

2) The paper is well written and is supported by good number of experiments.

**Weaknesses:**

1) While the paper motivates the ring topology as a means to improve communication efficiency and remove reliance on a central server, the justification for specifically adopting a ring structure remains somewhat unconvincing. For instance, the introduction mentions vehicle-to-vehicle networks as a potential application. However, in such dynamic environments, client participation is highly transient where vehicles continuously enter and leave the network. It is therefore unclear how the proposed framework would ensure stable participation long enough to complete a full and have several communication cycles around the ring. In practice, decentralized systems could still achieve low communication cost through peer-to-peer communication within dynamic local neighborhoods rather than enforcing a fixed ring structure. Clarifying why a strict ring topology is chosen over more flexible peer-to-peer connectivity, and how DRDFL handles client disconnection in such dynamic scenarios, would strengthen the motivation and applicability of the approach.

2) All reported experiments appear to assume 100% client participation in each communication round within the ring topology. It would be valuable to understand how DRDFL performs under partial client participation. For instance, with 50 total clients, one could randomly sample small subsets (e.g., 5 clients) to form temporary rings per round, with different subsets participating in subsequent iterations.

3) The paper evaluates DRDFL under moderate non-IID settings, but it remains unclear how the method would perform in extreme heterogeneity scenarios, specifically where each client possesses a distinct set of class labels (i.e., no label overlap).

4) How would DRDFL behave under a linear (path) topology, e.g., A → B → C → D, where no cycle exists? In a directed path, upstream clients (e.g., A) never receive updated Learngene or class statistics from downstream nodes, so consensus information cannot propagate back.

5) The evaluation is restricted to ResNet-18 and small datasets, with all clients participating every round. This leaves an open question, how DRDFL scales to larger models and datasets. Expand the study with larger models and datasets. For example use ViT-B/16 and TinyImageNet.

6) The proposed framework requires each client to communicate class statistics Learngene optimization. This will not have privacy implications behind sharing class distributions of clients?

7) For fairness, all baselines and DRDFL should be compared under same computation cost. The authors should report figures where the x-axis is the computation cost, and accuracy on the y-axis.

8) All experiments were performed under a homogeneous model architecture (ResNet-18) across clients, yet the comparison set omits decentralized model-averaging baselines that also use re-Weighted SoftMax (WSM) cross-entropy [1] during training mechanisms to mitigate data heterogeneity. Prior work such as DFML [2] have used such setup as a baseline. Compare DRDRL with WSM in decentralized FL setting.

[1] Legate et al. "Re-weighted softmax cross-entropy to control forgetting in federated learning". PMLR, 2023.

[2] Khalil et al. "DFML: Decentralized Federated Mutual Learning." TMLR 2024.

**Questions:**

Please address concerns in Weaknesses above.

---

> ### Author Response · Authors · 2025-11-21
> **Re: Response to Reviewer LsAK (Part 1)**
>
> Thank you very much for your thoughtful consideration and detailed analysis. We hope that the additional analysis and explanations will effectively address the concerns raised.
>
> > ### Q1:  Clarifying why a strict ring topology is chosen over more flexible peer-to-peer connectivity, and how DRDFL handles client disconnection in such dynamic scenarios, would strengthen the motivation and applicability of the approach.
>
> Please refer to the response to "CQ1 of Common Questions" for the reply.
>
> > ### Q2: It would be valuable to understand how DRDFL performs under partial client participation.
>
> We thank the reviewer for the valuable suggestion. Our experiments primarily adopt full client participation to evaluate convergence behavior in large-scale DFL settings without introducing additional variance from random sampling. This design choice is grounded in decentralized convergence theory, where a fundamental requirement for theoretical convergence is that the communication graph remains connected[1].
>
> we conducted an additional experiment under a fully connected DFL, which represents a robust special case of time varying or random graphs where global connectivity is always preserved [2]. In this topology, global connectivity is guaranteed even if only a subset of clients participate in each round. The results, presented below for CIFAR-10 ($s$=4), clearly show that DRDFL maintains its significant performance advantage over the DFedPGP.
> | Method| Local_T| Global_T
> | - | -|-
> | DFedPGP | 85.65%|32.07%
> | DRDFL| 88.36 %|35.25%
>
> [1] Shi W et al. Extra: An exact first-order algorithm for decentralized consensus optimization[J]. SIAM Journal on Optimization, 2015. [2] Nedić A et al. Distributed optimization over time-varying directed graphs[J]. IEEE Transactions on Automatic Control, 2014.
>
> > ### Q3: The paper's performance in extremely heterogeneous scenarios (i.e., no label overlap) remains unclear.
>
> The extreme heterogeneity scenario suggested by the reviewer, in which each client possesses a completely distinct set of class labels, is a strong structural assumption that can only occur when the number of clients is smaller than the total number of classes. To further investigate this case, we conducted an additional experiment on CIFAR-100 with 20 clients, where each client was assigned a unique, non-overlapping set of class labels. Each client achieves strong personalized performance, while generalization to the global distribution is significantly reduced.
> | Method    | Local_T   | Global_T  |
> | - | - | - |
> | DFedPGP   | 80.69%     | 4.02%      |
> |DRDFL | 86.82% | 5.98%|
>
> > ### Q4: How would DRDFL behave under a linear (path) topology, where no cycle exists?
>
> Thank you for your valuable feedback on linear (path) topology. We would like to clarify that the design and analysis of DRDFL are based on the standard assumption in decentralized optimization: the communication graph must be connected to ensure bidirectional information propagation. Directed path structures cannot support the reverse transmission of consensus information; this limitation stems from the topology itself, not the DRDFL methodology. This type of graph does not satisfy the fundamental premise of decentralized optimization and exceeds the basic setup of collaborative federated learning.
>
> >### Q5: This leaves an open question, how DRDFL scales to larger models and datasets. Expand the study with larger models and datasets. For example use ViT-B/16 and TinyImageNet.
>
> Thank you for pointing out this limitation. To further demonstrate the scalability of our approach, we have conducted additional experiments using a larger model and dataset, specifically ViT-B/16 on the TinyImageNet dataset under the setting ($s$ = 20). The results confirm that DRDFL consistently outperforms the strong baseline DFedPGP, showing that our method can generalize well to more complex architectures and datasets.
>
> | Method  |  Local_T  | Global_T |
> | -- | - | -|
> | DFedPGP  |   46.61%   |   4.15%   |
> | DRDFL  | 51.42% | 4.32% |

---

> ### Author Response · Authors · 2025-11-21
> **Re: Response to Reviewer LsAK (Part 2)**
>
> > ### Q6: The proposed framework requires each client to communicate class statistics Learngene optimization. This will not have privacy implications behind sharing class distributions of clients?
>
> Thank you for this important question regarding privacy implications. In DRDFL, each client exchanges global-level class statistics (i.e., one embedding per class). Every client transmits a complete set of class-wise statistics, and every client also receives a complete set from its neighbors. Importantly, a client only updates the statistics corresponding to its own classes, while the remaining class entries remain unchanged. **Since all transmitted messages have identical structure and dimensionality, no peer can infer which classes are actually present on a client**.
>
> >###  Q7: For fairness, all baselines and DRDFL should be compared under same computation cost. The authors should report figures where the x-axis is the computation cost, and accuracy on the y-axis.
>
> Thank you for the suggestion. We agree that fairness is of paramount importance in empirical comparisons. However, enforcing strictly equal computational costs across fundamentally different algorithms without altering their original designs is often impractical. Forcing such alignment may require modifying architectures or local training procedures, which would compromise the integrity of the methods and result in unfair or invalid comparisons.
>
> Following prior FL works, we adopt a standard evaluation protocol where all methods are trained under identical configurations: same batch size, number of local training epochs, and hardware environment. We adhere to this commonly accepted setting to ensure meaningful and consistent comparisons across approaches.
>
> >###  Q8:  Compare DRDRL with WSM in decentralized FL setting.
>
> We sincerely thank the reviewer for pointing out this relevant line of work. We agree that WSM cross-entropy represents meaningful baseline for mitigating data heterogeneity.  The objective of WSM-based approaches differs from ours. WSM aims to maintain a single global model and **mitigate catastrophic forgetting** across clients, often at the cost of personalization. Our method focuses on improving personalized inference while still capturing decentralized consensus knowledge.
> Nevertheless, we appreciate the reviewer’s suggestion and have included a comparison with a decentralized WSM (FedWSM) using the same ResNet18 architecture under CIFAR-100 with $s$=20. The results are shown below:
> | Method| Local_T| Global_T  |
> | --| -- | --|
> | FedWSM| 67.17%| **14.75%** |
> | **DRDFL** | **71.19%** | 13.40% |
>
> The WSM improves global prediction consistency but compromises personalization, whereas our method achieves significantly stronger personalized performance while maintaining comparable global accuracy. We will include a discussion of the relevant methods in the revised manuscript.

---

> > ### Comment · Reviewer_LsAK · 2025-11-27
> >
> > I would like to thank the authors for addressing my questions and comments.
> >
> > Regarding the author's comment that "fixed topology may be impractical", have you experimented with dynamically changing the topology during training? Additionally, a ring topology is not the only structure that enables minimal communication—one could also consider a star-like topology in which, at each iteration, only one client communicates with another.
> >
> > For questions 2, 3, 5, and 8, the reported results remain unconvincing. Key information is missing, including computation cost, convergence speed, and variance across multiple trials. Including these metrics is essential for evaluating the robustness and practical applicability of the proposed method.

---

> ### Author Response · Authors · 2025-12-02
>
> Thank you for your timely and thoughtful feedback, which gave us the opportunity to clarify the relevant issues.
>
> -   **Regarding our statement that “fixed topology may be impractical”**, there may have been some misunderstanding. Our intention is to clarify that our method can be naturally extended to other topologies and is not restricted to a fixed ring topology.
>
> -   **With respect to the comment on the “star-like topology”**, we would like to clarify that our definition of a star topology follows the conventional federated learning setting, where a central server communicates with multiple clients, as discussed in Appendix B.2.7. The scenario described by the reviewer is more closely related to the setting adopted in the DFML work. In our response to Q2, our setting is similar to partial client participation at each iteration, where one client communicates with other clients.
>
> -   **Regarding the computational cost**, the models and hardware used in the experiments reported in our responses are the same as those used in the submitted manuscript. The corresponding training time and floating-point operations (FLOPs) have already been reported in Appendix B.2.3.
>
> -   **For Q2, 3, 5, and 8**, the visualization of the convergence analysis is currently presented in Figure 12 of Appendix B.2.8. The reported results are based on three independent runs with different random seeds and are presented in the mean ± standard deviation format.
> ### Q2:
> | Method   | Local-T (mean ± std) | Global-T (mean ± std) |
> |----------|------------------------|-----------------------------|
> | **FEDPGP** | **85.20 ± 0.30**        | **32.92 ± 0.62**              |
> | **DRDFL**  | **89.14 ± 0.59**        | **35.58 ± 0.24**              |
>
>
> ### Q3:
>
> | Method   | Local-T (mean ± std) | Global-T (mean ± std) |
> |----------|------------------------|-----------------------------|
> | **FEDPGP** | **79.39 ± 1.17**        | **4.083 ± 0.045**            |
> | **DRDFL**  | **87.13 ± 1.58**        | **5.00 ± 0.84**              |
>
> ### Q5:
>
> | Method   | Local-T (mean ± std)   | Global-T (mean ± std) |
> |----------|--------------------------|-----------------------------|
> | **FEDPGP** | **46.81 ± 0.28**          | **4.14 ± 0.01**               |
> | **DRDFL**  | **51.36 ± 0.72**          | **4.69 ± 0.28**               |
>
> ### Q8:
>
> | Method   | Local-T (mean ± std)   | Global-T (mean ± std) |
> |----------|--------------------------|-----------------------------|
> | **FedWSM** | **67.45 ± 0.50**          | **14.39 ± 0.67**              |
> | **DRDFL**  | **71.67 ± 0.49**          | **13.93 ± 0.46**              |

---

### Official Review · Reviewer_E5sm · 2025-10-31

**Soundness:** 3
**Presentation:** 3
**Contribution:** 3
**Rating:** 4
**Confidence:** 4

**Summary:**

The paper proposes DRDFL, a novel framework for Decentralized Federated Learning (DFL) on a ring topology. The core idea is to tackle data heterogeneity by decoupling the model into two components: a personalized PersonaNet module and a consensus Learngene module. The PersonaNet is trained to capture client-specific feature distributions using a Gaussian mixture model. The Learngene, which is the only component communicated between clients, is trained to learn class-invariant, generalizable knowledge through an adversarial optimization process against a uniform label distribution. The authors claim that this "divide-and-conquer" approach achieves superior personalization and generalization performance with very low communication overhead (0.58M parameters) in a ring-topology setting.

**Strengths:**

1. The proposed architecture, which combines a VAE-like structure with a dual-branch encoder for personalization (PersonaNet) and generalization (Learngene), is novel in the context of DFL. The specific technique of using an adversarial classifier trained towards a uniform distribution to enforce class-invariance in the Learngene module is a clever and interesting contribution.

2. The paper addresses a highly relevant and challenging problem: efficient and effective DFL under sparse communication topologies (ring) and severe data heterogeneity. A method that can achieve strong performance with minimal communication cost would be a significant advancement for practical peer-to-peer applications like collaborative autonomous driving or edge IoT systems.

**Weaknesses:**

1. The proposed DRDFL framework is exceedingly complex. It involves a VAE-like structure with two separate encoders, a decoder, a primary classifier, and an adversarial classifier. The optimization involves minimizing a combination of at least seven different loss terms: reconstruction loss ($L_{rec}$), classification loss on original and noised data ($L_{ce}$), two losses for PersonaNet ($L_{cls}, L_{log}$), and three losses for Learngene ($L_{kl}, L_{adv}, L^u_{adv}$). This complexity raises serious questions about the method's practicality, stability, and sensitivity to hyperparameter tuning. The ablation study in Table 5 is insufficient as it only removes entire loss groups (L_PR or L_GL), failing to justify the necessity of each individual component.

2. The convergence analysis provided in Appendix A.2-A.3, while a good effort, feels generic and disconnected from the core novelties of the method. Specifically, Assumption 3, which bounds the parameter variation of the Learngene module ($||\tilde{\phi}^t - \phi^t_m||^2_2 \leq \delta^2$), is a very strong and potentially unrealistic assumption. In a highly non-IID setting, the Learngene received from a neighbor with a drastically different data distribution could be very far from the client's current version, making $\delta^2$ large and rendering the convergence bound vacuous. The analysis does not seem to capture the dynamics of the adversarial training or the VAE objective, which are central to the proposed method.

3. The paper champions the ring topology for its communication efficiency but largely ignores its critical drawbacks. The primary issue is the slow information mixing time, which is linear in the number of clients ($O(M)$). Information from one client takes $M-1$ rounds to reach its other neighbor. This could severely hamper convergence speed in large-scale networks (e.g., M > 100), a limitation not explored in the experiments (which use M=20 and M=50). More importantly, the paper dismisses the critical issue of node failure as "future work". In a ring, a single disconnected client breaks the entire communication loop, halting the training process. This is a major practical failure point that undermines the viability of the proposed approach for real-world systems.

**Questions:**

1. The system is a complex amalgamation of multiple techniques (VAE, GMM, adversarial learning, noisy reconstruction). Could the authors provide a more detailed ablation study to justify the inclusion of each component? For instance, what is the performance impact of removing the noisy reconstruction, or using only the KL-divergence for the Learngene without the adversarial component? How sensitive is the model to the relative weighting of the numerous loss terms?

2. Regarding Assumption 3 in your convergence proof, how do you justify that the norm difference between the received and local Learngene module is bounded by a small constant $\delta^2$, especially in the early stages of training or under extreme label skew where neighboring clients might have completely disjoint class sets?

---

> ### Author Response · Authors · 2025-11-21
> **Re: Response to Reviewer E5sm (Part 1)**
>
> Thank you for dedicating your time and expertise to review our manuscript. We deeply appreciate your thoughtful and constructive feedback.
>
> > ### W1&Q1:  The system's high complexity is not sufficiently justified by the presented ablation studies. A more detailed component-wise analysis is necessary to demonstrate the contribution of each part and to address concerns about stability and hyperparameter sensitivity.
>
> We thank the reviewer for this insightful observation. We would like to clarify that the goal of our design is to **structurally disentangle** the inherently intertwined objectives of generalization and personalization in FL into two interpretable and modular components: *PersonaNet* and *Learngene*. This structure enhances analytical clarity and interpretability. Conceptually, the overall framework consists of two logical groups: an extended VAE and a classifier.
>
> The reconstruction loss ($L_{rec}$ ) is the primary VAE loss, while the two sub-terms of variational term ($L_{{PR}}$), namely ($L_{{cls}}$) and ($L_{{log}}$), are derived from a unified variational objective and reformulated for ease of optimization. For the generalization component ($L_{{GL}}$), in addition to the standard variational term ($L_{{kl}}$), we design an **auxiliary adversarial loss that explicitly enforces cross-client generalization; this term can be independently ablated**.
>
> For the classifier module, the cross-entropy loss ($L_{{ce}}$) is complemented with noised data to encourage more robust and transferable decision boundaries; this component can also be ablated to isolate its contribution. Together, these losses arise from well-defined variational and classification objectives rather than seven unrelated optimization terms, and their grouping reflects the functional decomposition of the model rather than excessive design complexity.
>
> We conducted a ablation experiment on adversarial loss and noisy data with a $s=4$ setting on the CIFAR-10 dataset. The specific results are shown below:
>
> | Method    | Local_T   | Global_T  |
> | - | -| -|
> | DRDFL w/o adv|  91.57% |  34.57%|
> | DRDFL w/o noisy|  91.89% |  35.70%|
> | **DRDFL**|92.25% | 36.67%|
>
> For the relevant parameter sensitivity analysis, please refer to the response to "CQ2 of Common Questions" for the reply.
>
> >### W2:  Assumption 3 is very strong and potentially unrealistic, and the convergence bound may break down under highly non-IID settings.
>
> Thank you for pointing this out. Regarding Assumption 3, we clarify that it is a standard bounded-difference assumption widely used in the convergence analysis of federated and decentralized optimization. It is typically assumed that local models, gradients, or that their pairwise differences are upper-bounded by a constant. Our assumption on the norm difference between the received and local Learngene modules plays exactly the same role, but is instantiated at the level of the Learngene parameters.
>
> In DRDFL, all *Learngene* modules are pretrained from the same initialization before collaborative learning begins. During FL, they use bounded step sizes, perform a fixed number of local updates, and operate over a shared class space. These conditions keep their parameter trajectories within a compact and stable region of the parameter space. Thus, the deviation between local and received *Learngene* parameters is finite, though not required to be small or diminishing; the constant $\delta$ simply contributes to the steady-state error term, as in standard convergence analyses.
>
> Even under extremely non-IID conditions (e.g., no label overlap), we empirically observe that the differences between neighboring Learngenes remain bounded and that convergence is still achieved.

---

> ### Author Response · Authors · 2025-11-21
> **Re: Response to Reviewer E5sm (Part 2)**
>
> > ### W3: Further clarification is needed on the ring topology’s scalability and robustness, particularly regarding potential node failures.
>
> We appreciate the reviewer pointing out this point. We acknowledge that there has been no detailed discussion on the issue of node failure, and a simple ring network may become fragile if a single client disconnects. This is not the fundamental limitation of DRDFL, but an engineering problem faced by all decentralized communication systems. In real-world distributed systems, ring based communication generally has basic fault-tolerant mechanisms during deployment. We  can use a standard and simple solution based on a timeout fault detector:
>
> *if a node does not respond within a threshold time, its upstream node will bypass that node and directly connect to its subsequent nodes, effectively ensuring uninterrupted system training. This mechanism does not interfere with the update rules of DRDFL, maintains communication mode, and eliminates the risk of the entire system stalling.*
>
>  We have added this mechanism to lines 951-955 on page 18 of the revised manuscript. To further evaluate large-scale networks, we conducted an experiment with M=100 on the CIFAR-10 dataset, and the results are shown below:
>
> | Method| Local-T| Global-T
> | - | -- |--
> | DFedPGP | 74.95%|33.82%
> | **DRDFL**  | 77.13 %|35.25%
>
> > ### Q2: Clarification is needed on justifying the bounded difference in Learngene parameters under early-stage training or extreme label skew.
>
> Regarding Assumption 3, we appreciate the reviewer’s careful consideration of the extreme non-IID scenario. To further examine this case,  we conducted an additional experiment on CIFAR-100 with 20 clients, where each client was assigned a unique, non-overlapping set of class labels. Each client achieves strong personalized performance, while generalization to the global distribution is significantly reduced.
> | Method    | Local_T   | Global_T  |
> | - | - | - |
> | DFedPGP   | 80.69% | 4.02%|
> |DRDFL | 86.82% | 5.98%|
>
> These results show that even under **fully disjoint label partitions**, DRDFL still yields clear personalized performance gains over a strong decentralized baseline (DFedPGP), and training remains stable. In such extreme settings, the constant in Assumption 3 can indeed be larger, which is naturally reflected in a looser convergence bound. However, the empirical behavior indicates that the deviation between received and local *Learngene* modules remains bounded in practice rather than diverging, and the convergence analysis continues to provide a meaningful characterization of the algorithm’s behavior. We will add this challenging non-overlapping experiment and a clarification of the interpretation of Assumption 3 in the revised manuscript.

---

### Official Review · Reviewer_tDu8 · 2025-10-31

**Soundness:** 2
**Presentation:** 2
**Contribution:** 2
**Rating:** 4
**Confidence:** 3

**Summary:**

The work introduces DRDFL a framework for decentralized federated learning under ring topology communication. The method decomposes learning into two modules : i) Learngene that is trained adversarially under a uniform label distribution to capture global, class invariant knowledge helping with label skew; and ii) PersonaNet a personal network based on Gaussian mixture distributions to learn and preserve client specific features and alleviate feature distribution skew. Clients share Learngene parameters and Gaussian statistics with neighbors to enable peer to peer learning.

**Strengths:**

1. The separation of the personalization and generalization concerns by having two separate modules for the peer to peer learning setting is new.
2. The use of adversarial training with uniform priors for Learngene and Gaussian mixture modeling for PersonaNet seems to effectively bridge the personalized representation learning with federated optimization.
3. The experiments on multiple datasets show superior performance of the proposed algorithm.

**Weaknesses:**

1. While the idea is novel for the peer to peer learning network, similar ideas have existed in the FL setups with access to a central server.
2. The paper is somewhat hard to follow and the writing could better describe the intuition of each component and design choices, for example how the adversarial uniform constraints achieve global invariance, etc.
3. If and how these two modules interact with each other is under discussed.

**Questions:**

1. How sensitive is the method to the hyperparameters controlling the balance between generalization and personalization losses?

---

> ### Author Response · Authors · 2025-11-21
> **Re: Response to Reviewer tDu8**
>
> We sincerely thank you for your constructive and thoughtful comments. We greatly appreciate the time and effort you devoted to reviewing our manuscript.
>
> > ### W1: While the idea is novel for the peer to peer learning network, similar ideas have existed in the FL setups with access to a central server.
>
> We appreciate the reviewer’s insightful perspective. The study of personalization and generalization in FL has attracted significant attention in recent years. A natural and intuitive approach to address these challenges is to decouple the model into shared and personalized components. From 2021 to 2025, a number of methods have been proposed within centralized FL settings, where access to a central server is assumed. This server enables global aggregation and coordination, thereby facilitating techniques such as partial model sharing or disentangled representation learning.
>
> In contrast, our work departs from this centralized paradigm and tackles a more challenging and underexplored setting: decentralized FL. In this scenario, communication is restricted to immediate neighbors in a ring topology, which imposes much stricter constraints on communication bandwidth and significantly complicates the process of collaborative consensus-building. To the best of our knowledge, existing work rarely addresses this setting in a comprehensive manner, particularly in terms of jointly considering communication efficiency, privacy preservation, and label distribution heterogeneity. Our proposed approach is explicitly designed to address these challenges, offering a novel solution tailored for this critical yet insufficiently explored decentralized FL environment.
>
> > ### W2: The paper is somewhat hard to follow and the writing could better describe the intuition of each component and design choices, for example how the adversarial uniform constraints achieve global invariance, etc.
>
> Thank you for pointing this out. We have refined the explanations of the principles and design of each component. In particular, we agree with your observation that the rationale behind the adversarial uniformity design was insufficiently articulated. We have revised this part in the updated manuscript, specifically in the blue-highlighted text on Page 7, Lines 325–332, as follows:
>
> *To equip *Learngene* with strong generalization ability, our objective is to ensure that its latent representation $\mathbf{z}_l$ remains client-agnostic and unbiased with respect to class distributions. The key intuition is that, under label distribution skew heterogeneity, each client observes only a subset of classes with highly imbalanced frequencies. We explicitly regularize $\mathbf{z}_l$ by enforcing the output of an auxiliary adversarial classifier to follow a uniform label distribution. This uniformity constraint compels the module to discard class-dominant patterns unique to individual clients and instead retain only the invariant, globally shared knowledge of the data.* In addition, we provide empirical visualizations to further support this mechanism. As shown in Figure 4 of Section 5.2 and Figure 9 in Section B.2.5, our method effectively captures class-level global invariances in a clear and intuitive manner.
>
> > ### W3: If and how these two modules interact with each other is under discussed.
>
> Thank you for raising this important point. The two modules in our framework are designed using a divide-and-conquer principle, each focusing on a distinct aspect of the representation (global invariance vs. personalization).  The *Learngene* module employs adversarial learning to approximate a uniform distribution, which mitigates label distribution skew across clients. In contrast, the *PersonaNet* module models global class representations to alleviate feature distribution skew. Each module therefore targets a distinct source of heterogeneity in decentralized federated learning.
>
> The outputs of the two modules are concatenated in the latent space and jointly consumed by the decoder. Through reconstruction, the decoder implicitly enforces compatibility between the two representations, ensuring that they form a coherent generative space. This coupling allows the reconstructed data to incorporate both globally shared patterns and client-specific traits, thereby achieving a unified integration of generalization and personalization.
>
> > ### Q1: How sensitive is the method to the hyperparameters controlling the balance between generalization and personalization losses?
>
> Please refer to the response to "CQ2 of Common Questions" for the reply.

---

### Official Review · Reviewer_kvHE · 2025-11-01

**Soundness:** 4
**Presentation:** 3
**Contribution:** 3
**Rating:** 6
**Confidence:** 3

**Summary:**

In this paper, the authors present DRDFL, a method for personalized, decentralized federated learning that aims to improve personal model performance via two modules. The paper decomposes representation learning into a private PersonaNet module to fix feature skew and a shared, adversarially-trained Learngene module to fix label skew. Their latents are combined for classification, while only the Learngene module and global Gaussian means and variances are communicated between clients. The paper compares their method against various SOTA baselines in the literature and across different heterogeneity measures. In terms of the paper’s strengths, they demonstrate that their method improves on or is comparable to the accuracy of other methods in most experiments. Furthermore, their method achieves these improvements with a much lower communication overhead between clients. However, the paper would benefit from a clearer motivation and definition of the personalized FL setting, as well as the clarification of minor details in the questions section.

**Strengths:**

* Overall, the paper is well-written, thorough, and generally persuasive.


* The proposed method demonstrates superior or comparable performance to existing baselines from the literature when evaluated on both global and local test accuracy across a range of heterogeneity conditions.


* The approach achieves roughly two orders of magnitude lower communication cost in terms of transmitted parameters compared to other centralized and decentralized federated learning methods.


* The authors also provide very extensive ablation studies and supplementary analyses that answered most questions that I had about the work. These include
   * Alternative communication topologies
   * Computational overhead of DRDFL
   * Convergence behavior
   * Robustness under gradient reconstruction attacks
   * Effects of removing individual loss components
   * Performance with an increased number of clients
   * Scalability to newly joined clients

**Weaknesses:**

* The first claimed contribution of the paper is the delineation of heterogeneity into feature skew and label skew. However, this distinction is already well-established in the federated learning literature (e.g., [1]) and therefore should not be presented as a novel contribution.


* The motivation for the personalized DFL objective is also insufficiently articulated. Additionally, it is unclear whether the Dirichlet-$\beta$ and shard-based partitioning schemes used in the experiments meaningfully represent realistic personalization scenarios. From an intuitive perspective, assessing model personalization with the dirichlet partition over labels would encourage client models to learn solutions heavily biased in favor of the local label skew with poor generalization quality. While this limitation may reflect broader challenges within the personalized FL subfield and extend beyond the scope of this particular work, the paper would nonetheless benefit from a more explicit discussion of these conceptual issues.

**Questions:**

* The personalization objective should be defined more clearly and introduced earlier in the paper. This objective represents a key distinction from works such as DFedAvg, where the aim is to jointly train a single global model. While the authors implicitly reference this objective through the use of Local-T and Global-T curves, it would strengthen the manuscript to articulate it explicitly in the early sections.


* Instead of reporting “0.58 M of communication” as a standalone statistic in the abstract and contributions section, it would be more informative to contextualize this value by comparing it to state-of-the-art baselines from the start (e.g., indicating that the method transmits an order of magnitude fewer parameters). The communication cost is relative to the size of the backbone classifier, so the sole number is not very explanative on its own.


* The manuscript should specify which backbone classifier is used across all methods, as this detail is necessary for interpreting performance and communication comparisons.


* The qualitative analysis using Grad-CAM is an interesting direction. However, to substantiate the claims about what is learned by PersonaNet and Learngene, the authors should include at least a few more visual examples in the Appendix to better justify their interpretations. It is not clear whether the image presented is representative of the majority of the images.


* Why is the work limited to the ring topology? It is unclear why the method cannot be used on any DFL topology.


References

[1] J. Pei, W. Liu, J. Li, L. Wang and C. Liu, "A Review of Federated Learning Methods in Heterogeneous Scenarios," in IEEE Transactions on Consumer Electronics, vol. 70, no. 3, pp. 5983-5999, Aug. 2024.

---

> ### Author Response · Authors · 2025-11-21
> **Re: Response to Reviewer kvHE**
>
> Thank you very much for taking the time and effort to review our manuscript. We sincerely appreciate your insightful comments.
>
> > ### W1: Clarification needed on the claimed novelty of separating feature skew from label skew.
>
> We sincerely thank the reviewer for the helpful comment. We acknowledge that feature distribution skew and label distribution skew are fundamental problems in federated learning. Our intention is not to claim these concepts as novel. Rather, we note that in decentralized federated learning, existing work has predominantly focused on addressing feature distribution skew within clients to improve personalization, while the label skew that simultaneously exists across clients has received comparatively less attention.
>
> Furthermore, the absence of a central server to coordinate model updates, combined with limited client-to-client communication, magnifies the adverse effects of data heterogeneity. For this reason, our manuscript provides a clear mathematical analysis of both forms of skew and proposes DRDFL to mitigate their amplified impact in decentralized settings.
>
> > ### W2: The paper lacks a clear motivation for the personalized DFL objective and uses partitioning schemes (Dirichlet- $\beta$  and sharding) whose realism for evaluating personalization is questionable.
>
> Thank you for providing these valuable suggestions. The motivation behind our proposed method is that existing DFL methods primarily focus on the issue of feature distribution skew, yet they often overlook the label distribution skew across clients. Our designed DRDFL aims to address this shortcoming, and the ultimate results reflect improvements in both the personalized and generalizable performance of the model.
>
> The two data partitioning strategies we adopted are among the most widely used in the federated learning literature, and these configurations align with established methodologies. Regarding the reviewer’s observation that Dirichlet partitioning may lead local models to overfit their local labels and exhibit poor generalization, we fully agree this fact. It is precisely why we emphasize both personalization and generalization capabilities of the model, and have introduced two metrics to quantitatively evaluate these aspects.
>
> > ### Q1: The personalization objective should be defined more clearly and introduced earlier in the paper.
>
> Thank you for your detailed feedback on our manuscript. Our proposed method aims to improve generalization performance across global data distributions while focusing on the personalized performance of the data distribution itself. We have clarified this point in the revised manuscript on Page 2, Lines 94–99, as follows:
>
> *As illustrated in Figure 1(b), methods such as Local and DFedPGP achieve strong personalized performance (Local-T) yet exhibit limited generalization capability (Global-T) to the global data distribution. In contrast, our approach enhances the model’s generalization ability toward the global data distribution while also influencing its personalized performance. This underscores the importance of jointly addressing both types of distribution skew in limited-communication RDFL, as well as the necessity of learning generalized and effective consensus knowledge.* Additionally, we have explicitly specified the optimization objective for each client in Equation (1).
>
> >### Q2:  The statistical data “communication cost of 0.58 M” is not clearly described.
>
> Thank you for your detailed suggestions. We have revised the corresponding content in the abstract on Page 1 (Lines 30–31) and in the contributions section on Page 3. The specific modifications are as follows:
>
> *Extensive experiments show that our method achieves superior performance in RDFL while reducing the communication cost to only 0.58 M, which is more than two orders of magnitude lower than the state-of-the-art baseline. This substantial reduction highlights the effectiveness of our approach in addressing data heterogeneity under stringent communication constraints.*
>
> >### Q3: The manuscript should specify which backbone classifier is used across all methods.
>
> Thank you for the detailed feedback. In the revised manuscript, we have explicitly specified the classifier used on Line 380 of the main text, rather than mentioning it only in the appendix.
>
> > ### Q4: The qualitative analysis using Grad-CAM is an interesting direction. The authors should add at least some visualization examples in the appendix to better support their interpretations.
>
> We sincerely thank the reviewers for their insightful comments regarding the need for more qualitative evidence. We have added a new Grad-CAM visualization and analysis to Appendix B.2.5 on page 23, lines 1203-1224 of the revised manuscript.
>
> >### Q5: Why is the work limited to the ring topology? It is unclear why the method cannot be used on any DFL topology.
>
> Please refer to the response to "CQ1 of Common Questions" for the reply.

---

> > ### Comment · Reviewer_kvHE · 2025-11-25
> >
> > We appreciate the authors’ responses to our comments regarding the presentation in Q2–Q4, as well as the clarification about the ring topology in CQ1. However, we reiterate that label skew is a well-studied phenomenon in both federated and decentralized federated learning (e.g., Dirichlet-based label partitioning, which is standard in much of the FL+DFL literature). Therefore, the statement that “existing work has predominantly focused on addressing feature distribution skew within clients” is not accurate. While the problem formulation is clear, this point should not be framed as a contribution. For this reason, we maintain our original rating of 6.

---

> > > ### Author Response · Authors · 2025-11-26
> > >
> > > We sincerely appreciate the time you have dedicated to reviewing our work and are grateful for your constructive feedback on our responses.
> > >
> > > We fully agree with your point that label skew is a well-studied phenomenon in both federated and decentralized federated learning, and that Dirichlet-based label partitioning is the standard approach for simulating data heterogeneity scenarios. We apologize for the wording in our original submission that may have caused misunderstanding—we did not intend to imply that Dirichlet-based partitioning only reflects label skew.
> > >
> > > What we intended to emphasize is the following:  **both label distribution skew and feature distribution skew are fundamental forms of data heterogeneity in FL**, and in decentralized settings, the limited communication budget can further exacerbate these challenges. While existing decentralized FL methods often focus on mitigating feature distribution skew to enhance client-side personalization, they typically place less emphasis on improving global generalization across all data distributions, especially under strict communication constraints. **In other words: feature distribution skew predominantly affects personalization, whereas label distribution skew influences generalization.**
> > >
> > > Our overarching objective is therefore to learn communication-efficient consensus knowledge that supports both generalization and personalization under limited communication. In the revised manuscript, we now describe this aspect within the broader context of our contributions as follows:
> > >
> > > *We revisit data heterogeneity in RDFL, where limited communication and the absence of a central coordinator amplify its impact, and reveal the importance of simultaneously considering both distribution skewness issues and the necessity of training generalized consensus knowledge.*
> > >
> > > We would greatly appreciate your involvement. Your feedback is instrumental in refining our work, and your contributions would be greatly appreciated. ﻿ Thank you for your time, dedication, and invaluable support.

---

### Author Response · Authors · 2025-11-21
**Common Questions**

> CQ1:  Reasons for choosing ring topology.

Thank you for the insightful comment. We would like to emphasize that the proposed methodology is not intrinsically tied to a ring structure. As detailed in Appendix B.2.7, we have explicitly extended and validated our approach under alternative and dynamically varying topologies. The experimental results confirm that our method remains effective and feasible across these more flexible connectivity patterns, addressing the concern that a fixed topology may be impractical.

The primary reason for presenting our method within a ring topology in the main manuscript is its simplicity and representational clarity. As one of the most basic connected topologies, the ring provides a stringent and transparent testbed for evaluating the core innovation of our work: the efficient propagation and consolidation of “consensus knowledge’’ under minimal communication. Demonstrating strong performance in such a sparse and high-diameter topology offers compelling evidence that the method can generalize and even perform better in richer peer-to-peer connectivity settings.

Moreover, as discussed in Appendix B.2.6, DRDFL naturally supports newly joined clients through an efficient initialization mechanism, ensuring seamless participation even under client churn. Likewise, transient disconnections do not disrupt the algorithm, as the *Learngene* is inherited only from direct neighbors rather than relying on the entire ring being intact. In practical scenarios such as vehicle-to-vehicle networks, the system can exploit latency buffers to detect departures and reconfigure the ring among active participants, enabling continuous operation despite dynamic membership.

> CQ2: How sensitive is the method to the hyperparameters controlling the balance between generalization and personalization losses?

Thank you for raising this important point. We would like to clarify that the generalization loss and personalization loss are both fundamental components of our framework, and preliminary experiments showed that they operate at similar scales. Therefore, we adopted the same default hyperparameter values in the main experiments.

To empirically assess the sensitivity, we conducted a 3×3 grid search over the two loss weights $L_{{GL}} \in$ $[0.5, 1, 2]$ and $L_{{PR}}$ $\in [0.5, 1, 2]$ under the CIFAR-100 dataset with ( $\beta$ = 0.1). Across all nine combinations, the performance remains highly stable, ranging from 91.42% to 92.86%, a fluctuation of **less than 1.5 percentage points**. This demonstrates that DRDFL is not sensitive to the precise choice of these hyperparameters, and a single default setting ($L_{GL} = L_{PR} =1$ ​) already achieves near-optimal performance without additional tuning. We will incorporate these results into the revised manuscript to ensure completeness and clarity.

| $L_{GL}$ / $L_{PR}$ | 0.5  | 1|2
| --------- | --------- | --------- |--------- |
| 0.5|  91.71%| 92.09%|91.57%
| 1|  92.77%|  92.86%|91.80%
| 2|91.89%| 92.38%|91.42%

---

### Note · Authors · 2026-01-26

I have read and agree with the venue's withdrawal policy on behalf of myself and my co-authors.

---

### Meta-Review · Area_Chair_Vtci · 2026-01-01

**Summary:**

This paper proposes **DRDFL**, a divide-and-conquer framework for ring-topology decentralized federated learning that separates personalized (PersonaNet) and invariant (Learngene) knowledge to address feature and label distribution skew. Reviewers generally found the paper to be well-written, technically solid, and empirically thorough, with particularly strong results on communication efficiency and extensive ablation studies.

However, overall reviewer support was limited and cautious. A major concern is the reliance on a (largely static) ring topology. As highlighted by Reviewer E5sm, ring topologies suffer from linear information mixing time, slow propagation in large networks, and fragility to node failure. These issues are serious for real-world deployment and are not fully explored in the main experiments, which focus on relatively small client counts.

Another recurring concern is novelty framing. The distinction between feature distribution skew and label distribution skew is already well-established in the FL literature and should not be presented as a core novel contribution. Reviewers felt the contribution is better viewed as an engineering solution for decentralized settings rather than a new conceptual insight.

### Rebuttal Assessment

The rebuttal addressed several points effectively. The authors clarified that DRDFL is not inherently tied to a strict ring topology and provided additional experiments on alternative and dynamic topologies, as well as clarifications on communication cost, backbone models, hyperparameter sensitivity, and qualitative analysis. They also reported added experiments on larger client counts and a fault-handling mechanism, which partially mitigates scalability and failure concerns.

However, key issues remain only partially resolved. The fundamental drawbacks of ring topologies (mixing time and robustness) are still central, and much of the justification relies on appendices and rebuttal text rather than the main narrative. The novelty concern regarding skew decomposition also remains.

### Recommendation

**Reject**

While the paper is technically and empirically strong, the combination of limited reviewer enthusiasm,topology-related practical concerns, and overstated novelty makes acceptance at ICLR challenging. That said, I believe this is still a solid piece of work, and with clearer positioning and stronger emphasis on adaptive or non-ring topologies in the main paper, it could be competitive in a future submission.

**Reviewer Concerns:**

See above.

**Reviewer Scores:**

* **Reviewer kvHE**: Likely unchanged (≈6), explicitly maintained due to novelty framing.
* **Reviewer LsAK**: Possibly slightly more positive after rebuttal, but likely still borderline.
* **Reviewer E5sm**: Unclear; major topology concerns may still prevent a higher score.
* **Reviewer tDu8**: Insufficient follow-up; likely unchanged.

Overall, reviewer sentiment would likely remain **mixed**.

---

### Decision · Program_Chairs · 2026-01-26

Reject